Subject Areas:
engineering geology/energy

Keywords:
sustainable development, coal mine, roof cutting, chain arm, retain roadway

Author for correspondence:
Yang Tai
e-mail: cumtcqty@163.com

# The sustainable development of coal mines by new cutting roof technology

Bin Yu[1], Yang Tai[2], Rui Gao[3], QiangLing Yao[2], Zhao Li[1] and Hongchun Xia[4]

[1]Science and Technology Center, Datong Coal Mine Group Co. Ltd, Datong 037000, People's Republic of China
[2]School of Mines, China University of Mining and Technology, Xuzhou 221116, People's Republic of China
[3]College of Mining Engineering, Taiyuan University of Technology, Shanxi 030024, People's Republic of China
[4]College of Architectural Engineering, Dalian University, Liaoning, Dalian 116622, People's Republic of China

YT, 0000-0003-0801-6261; RG, 0000-0003-1988-8778

China consumes more than 3.6 billion tons of coal every year. In the meanwhile, coal accounts for over 60% of the energy consumption sector. Therefore, the sustainable development of coal mines is a problem that needed to be solved by the Chinese government. During the coal resources recovery process, the protective coal pillars between the adjacent working faces lead to a vast waste of coal resources. In order to mitigate the resource-wasting issue, a new technology of roof cutting with chain arm retaining roadway was put forward in this paper. First, the procedures of retaining roadway, roof-cutting parameters and the damage ranges of roadway surrounding rock induced by roof cutting with chain arm were analysed. Then, the working resistance of the temporary support equipment is given when using the new technology to retain the roadway. Next, the roof-cutting height, the temporary support equipment selection, working resistance of portal support and support parameters of the bolt and anchor cables were optimized based on the numerical calculation. The industrial experiment of retaining roadway by roof cutting with chain arm was carried out in a working face. The surrounding rock damage was lowered and controlled with the application of chain arm roof-cutting technology. Also, it was found that the variation range of the uniaxial compressive strength was only 5%, resulting in the roof damage range of 82 mm. The new technology has proved a potentially wide application in the coal mining industry with prosperous economic and safety improvement.

**Figure 1.** Layout method of coal pillars. (*a*) Position of the roadway; (*b*) stress state of the roadway.

# 1. Introduction

China is the largest coal consumer in the world, and its coal consumption accounts for more than 60% of primary energy consumption [1]. However, with the high strength exploitation for past decades, coal resources are quickly exhausted, especially in China's eastern provinces, such as Jiangsu, Henan and Shandong. [2]. During the process of coal resources recovery, the protective coal pillars reserved to isolate gob for safe production for the next adjacent working faces have been a major cause for a low recovery rate [3]. Based on the width of the coal pillars, the design method of protective coal pillars can be divided into the large-pillar method and small-pillar method [4,5]. As shown in figure 1, the large-pillar method aims to arrange the roadway in the *in situ* stress area to avoid impacts of abutment pressure. This method is beneficial to maintain the roadway, but it will waste quite a large portion of coal resources [6]. In order to reduce the size of coal pillars and improve the coal recovery rate in coal mines, the small-pillar method has been developed. It is designed to arrange the roadway in the stress reduction area of the surrounding rock. The width of small coal pillars is generally between 3 and 6 m [6]. By applying this method, the roadway will locate in the stress reduction area, which makes the maintenance of the roadway easier and increases the recovery rate of coal resources [7]. Thus, the small-pillar method has become a priority for the roadway arrangement [8,9]. However, the advancing part for the roadway should be increased, because the small coal pillars are arranged in the stress reduction area, and the roadway surrounding rocks are in the plastic area [10,11]. Besides, the small-pillar method can only be carried out after the strata behaviour of the upper working face becomes stable, which may decrease the efficiency for preparing the next working face [12,13]. At the same time, the roadway along the gob is in the plastic failure area of the surrounding rock, as shown in figure 2. When thick-hard strata are located above the working face, the damage of coal pillars will

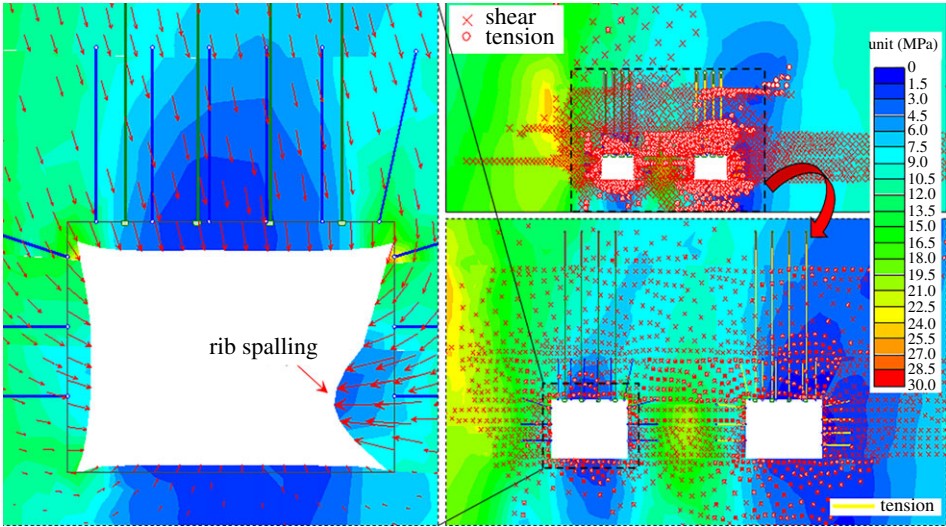

**Figure 2.** The roadway failure areas in the small-pillar method.

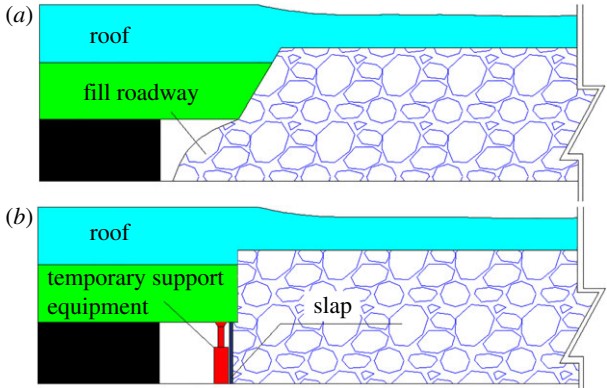

**Figure 3.** The treatment methods for roadway. (*a*) The traditional method; (*b*) non-pillar method.

be more serious. Due to the dynamic pressure of the thick-hard strata, the coal rib spalling often occurs near gob [14].

In order to increase the coal recovery rate, realize sustainable development for mines and improve the working efficiency for moving to the next working face in time, the non-pillar mining method has been adopted in coal production. In traditional mining methods, the roadway roof and gob roof are regarded as a whole structure, and their movements were highly correlated. The roof breaks and fills the roadway under the actions of mining pressure, as shown in figure 3*a*. Compared with the traditional roadway treatment methods, the non-pillar method will retain the gob-side roadway through cutting the roof method, which includes drilling hole and installing explosives in the roadway roof near gob and then implementing roof pre-splitting by directional energizing blasting technology. As shown in figure 3*b*, after the working face passes, the slap to prevent gangue should be timely adopted to avoid the slip of broken roof into the roadway. Meanwhile, the broken roof is crushed into the gob under the actions of surrounding rock pressure in the stope and the temporary supporting equipment. Under the influence of the expansion of rock mass, the broken roof fills the gob area and provides a positive supporting force to balance and stabilize the strata in the gob. The broken roof could support the overburden, control the subsidence to some degree and reduce the loads from cantilever beam structure to the coal pillar, which can guarantee the roadway stability. The collapsed roof formed one wall for the roadway, and the preserved roadway will service for the next mining of the working face.

At present, many studies on the non-pillar method have been carried out. He *et al*. [15,16] proposed the non-pillar method and developed the constant resistance and large-deformation anchor cable for this technology. The mine pressure law of the non-pillar working face was revealed. Besides, the working resistance for temporary support equipment was given, and the theory and technology of the

non-pillar method were established. Gao *et al.* [17,18] studied the dynamic impact behaviours of the gangue body under different mining heights. The mechanism and control techniques for gangue rib deformations were explored in detail. A new control approach was developed to solve the instability problem of the gangue rib in thick coal seams. Sun *et al.* [19] studied critical parameters in the non-pillar method to retain the gob-side roadway for thin coal seams mining. The roof-cutting height, pre-splitting angle, and the distance between each pre-splitting blasting hole were determined and optimized. Guo *et al.* [20] gave the theoretical formulae for the roof-cutting height and pre-splitting angle based on theoretical calculation.

These studies, as mentioned above, mainly focus on the non-pillar mining method to retain the gob-side roadway. Furthermore, the energizing blasting method is usually adopted as the roof-cutting method. It has a complicated operation process in drilling holes, installing explosives and detonating explosives. It also has a low mechanization degree and considerable disturbance damage to the roof. The broken roof has a poor self-standing ability. Besides, the directional blasting is hard to be accurate, and fracture between adjacent boreholes could not be ensured for penetration. In some extreme conditions like a high gassy coal mine, any blasting actions are prohibited. Aiming at the shortcomings of the energizing blasting, a weak-disturbance, high-efficiency, accurate and continuous roof-cutting method is in urgent need. Datong Coal Mine Group Co., Ltd has developed the technology and roof-cutting equipment with the chain arm to retain roadway. The new technology could effectively address the problems above, improve the recovery rate of coal resources, prolong the service life of mines and ensure their sustainable development.

The structure characteristics of the roof-cutting equipment with the chain arm and the process of roof cutting were firstly introduced in the paper. Next, the process of retaining roadways, roof-cutting parameters and damage range was analysed. Then, the critical technical parameters for the new technology were optimized by numerical calculation. Finally, the industrial experiment of hard roof cutting was carried out in Datong Coal Mine Group Co., Ltd.

# 2. The equipment of roof cutting with chain arm and technology of retaining roadway

The roof cutting with chain arm to retain roadway is to continuously and accurately cut the roof with weak disturbance and high efficiency by the chain arm assisted with a laser calibration system. It could provide the basis for retaining the gob-side roadway and improve the recovery rate of coal resources. The roof-cutting equipment and technology of retaining the roadway were mainly introduced in this section.

## 2.1. The equipment of roof cutting with chain arm

The vital equipment is the machine of roof cutting with a chain arm. As shown in figure 4, its components include chain arm, crawler travel mechanism, fixed mechanism, power unit and spray system. It has the following advantages: (i) no water needed; (ii) low-speed rotate and not easy to produce sparks; (iii) without pre-drilling and saving auxiliary time; (iv) smooth cutting surface; (v) continuous cutting, high-efficiency, low labour intensity and labour-saving operation; (vi) using the round polycrystalline diamond; (vii) usually no less than 1000 h service life; (viii) effective dust control via internal and external spray system; and (ix) effective and accurate tool setting by laser calibration system.

## 2.2. The process of roof cutting with chain arm

The roof-cutting process is illustrated in figure 5, and the procedures are (i) to start the crawler travel mechanism and place the machine at the end of the roadway; (ii) to use the crawler travel mechanism to make the chain arm close to coal wall of the working face, and keep the machine body parallel to the roadway; (iii) to use the oil cylinder of the fixed mechanism to maintain a distance of 10 to 20 mm above the ground; (iv) to start the power unit and drive the rotation of chain, and rotate the chain arm from one side to the other side at a constant speed; (v) to rapidly rotate to the initial horizontal position after moving to the other side; (vi) to adopt the crawler travel mechanism to make the continuous cutting machine move forward a certain distance; (vii) to use laser calibration system to

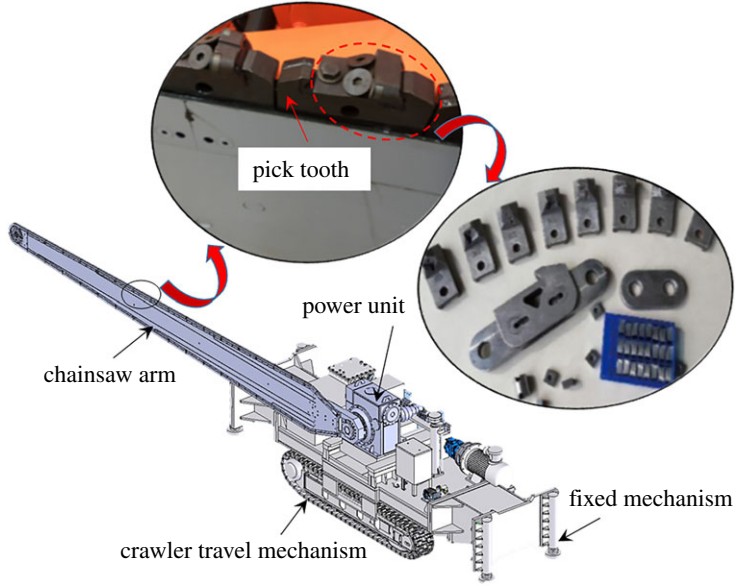

**Figure 4.** The machine of roof cutting with chain arm.

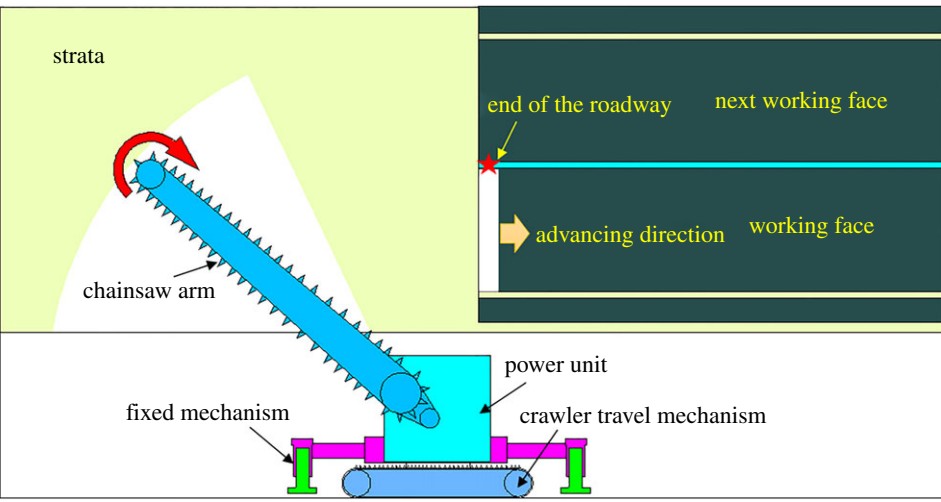

**Figure 5.** The process of roof cutting with chain arm.

keep the chain arm accurately align with the above cutting crack; (viii) to repeat the process of (ii) to (vi) and begin the next roof cutting.

# 3. The technology of roof cutting with chain arm to retain roadway

## 3.1. The process of retaining roadway

The technology could be divided into three steps, as shown in figure 6. Step 1: The machine of roof cutting with chain arm is used to cut the roof. Step 2: After the working face passes, temporary supports in the roadway should be strengthened, and the gangues should be prevented timely at the rear of the working face. The special unit support or portal support is mainly used as the temporary support. The slap is used to prevent a broken roof into the roadway. Step 3: The broken roof in gob is gradually compacted as the working face advances. The temporary support equipment can be removed after roof stabilization. The sprayed concrete is used for the rib formed by the broken roof to prevent air leakage. Finally, the process is completed.

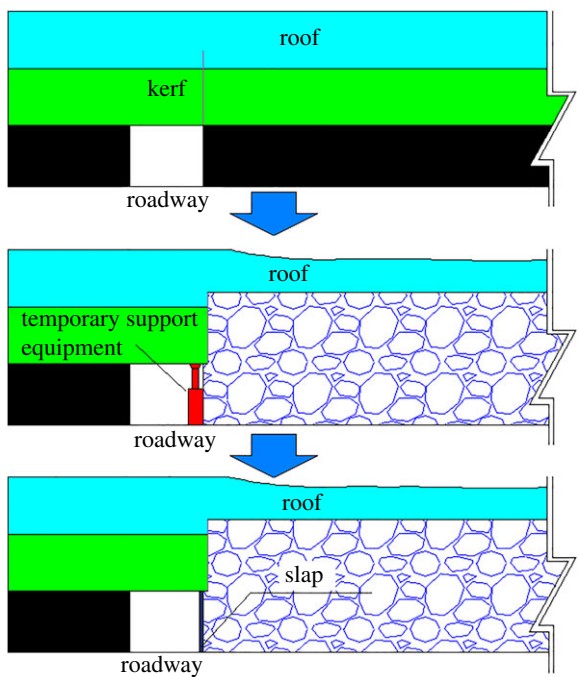

**Figure 6.** The process of retaining roadway.

**Table 1.** Coefficients for the caving zone height.

| type of the immediate roof | uniaxial compressive strength (MPa) | coefficients | |
| --- | --- | --- | --- |
| | | $c_1$ | $c_2$ |
| hard | >40 | 2.1 | 16 |
| mid-hard | 20–40 | 4.7 | 19 |
| weak | <20 | 6.2 | 32 |

## 3.2. Key technical parameters

In the technology of roof cutting with the chain arm to retain roadway, the technical parameters are essential to guarantee the effects of roadway retaining. They include the roof-cutting height, the type and working resistance for temporary support and support parameters of bolt and cable. This section mainly analysed the roof-cutting height and working resistance of the temporary support equipment.

### 3.2.1. Roof-cutting height

The roof-cutting height $H$ refers to the maximum vertical height of the kerf. In general, to guarantee that the broken roof could backfill the gob and ensure the stability of the strata, the height $H$ should be more than the caving zone $H_0$. The scholars have obtained the regression formula for the height of the caving zone through analysing the caving zone ranges in a large number of mines with different geological conditions in China and the USA, as follows [21]:

$$H_0 = 100h/(c_1h + c_2), \tag{3.1}$$

where $h$ is the mining height, m; $c_1$ and $c_2$ are parameters related to roof lithology, as shown in table 1.

Therefore, the roof-cutting height should satisfy: $H > 100h/(c_1h + c_2)$.

### 3.2.2. Working resistance of the temporary support equipment

There are three broken positions in the roof of the large-pillar method and small-pillar method, which are outside of the coal pillar, directly above the roadway and inside of the solid coal wall. The influencing factors include strata thickness, strata mechanical properties, mining depth, *in situ* rock stress state,

mining height, etc. Compared with the large-pillar method and small-pillar method, when roof cutting with chain arm technology is applied, the broken position of the roof is at the boundary of the gob. He et al. [22] have given the working resistance for the temporary support equipment, as shown in table 2.

## 3.3. The disturbance ranges of roof cutting to the roadway roof

Compared with roof cutting by the energizing blasting, the technology of roof cutting with chain arm could reduce the damage to the roadway roof. The rock damage range of the energizing blasting is shown below,

$$R_s = r_b \left[ \frac{\lambda P_b}{(1 - D_0)\sigma_t + p} \right]^{1/\alpha}, \tag{3.2}$$

where $r_b$ is the radius of the bore; $\lambda$ is the coefficient of lateral pressure, $\lambda = \mu/(1 - \mu)$; $\mu$ is the dynamic Poisson's ratio of the roof rock mass; $D_0$ is the initial damage of the rock mass; $\sigma_t$ is the tensile strength of the roof rock mass; $p$ is the *in situ* rock stress; $\alpha$ is the attenuation index of the explosive stress wave in the rock mass, $\alpha = 2 - \mu/(1 - \mu)$, which is related to roof lithological characters and blasting method; $P_b$ is the peak pressure of the shock wave in the pore wall.

To achieve the penetration effect between boreholes and control the damage range of the surrounding rocks from the blasting, the distance between adjacent boreholes is generally between 500 and 1000 mm. Therefore, the range of rock damage area by the traditional energizing blasting is between 500 and 1000 mm.

For the technology of roof cutting with chain arm to retain roadway, the rock damage range is usually less than 0.5 times the kerf width, which is usually 42 mm, so the damage range for the rock is less than 84 mm [23,24]. The technology of roof cutting with chain arm, by contrast, has smaller damage range to the roadway roof and is more conducive to the maintenance of the roadway roof.

# 4. The optimization for key parameters of the technology of roof cutting with chain arm to retain roadway

The key parameters include the roof-cutting height, the type and working resistance of the temporary support equipment, and the support parameters of bolt and anchor cable. They were optimized by the numerical simulation method to ensure the effects of the roof cutting with chain arm to retain the roadway.

## 4.1. The numerical model

### 4.1.1. Rock mass strength criterion at the stope

Due to a large number of irregular joints and cracks in the rock mass, the rock parameters obtained from the laboratory are usually higher than those at the stope. To simulate the strength reduction of the coal and rock mass, Hoek & Bray proposed the Hoek–Brown failure criterion in 1984, which was revised as the generalized Hoek–Brown criterion [25,26] and could be expressed as follows:

$$\sigma_1 = \sigma_3 + \sigma_{ci} \left( m_b \frac{\sigma_3}{\sigma_{ci}} + s \right)^a, \tag{4.1}$$

where $\sigma_1$ is the maximum principal stress under damage; $\sigma_3$ is the minimum principal stress; $m_b$ is a reduced value (for the rock mass) of the material constant $m_i$ (for the intact rock); $s$ and $a$ are constants which depend upon the characteristics of the rock mass; $\sigma_{ci}$ is the uniaxial compressive strength (UCS) of the intact rock pieces. The expressions of $m_b$, $s$ and $a$ are as follows:

$$m_b = m_i \exp\left( \frac{GSI - 100}{28 - 14D} \right), \tag{4.2}$$

$$s = \exp\left( \frac{GSI - 100}{9 - 3D} \right) \tag{4.3}$$

and

$$a = \frac{1}{2} + \frac{1}{6}(e^{-GSI/15} - e^{-20/3}), \tag{4.4}$$

**Table 2.** Working resistance of the temporary support equipment [22].

| broken characteristics of the strata | mechanical model | working resistance |
|---|---|---|
|  |  | $\begin{aligned} P = &[M_B(F_G/K_G + x_0 + a + b)K_G/F_G + q(F_G/K_G)^2 \\ &+ q(x_0 + a + b)^2/2 + qF_G(x_0 + a + b)/2/K_G \\ &+ q_0(x_0 + a + b)^2/2 - F_G(x_0 + a + b) \\ &- qL(h_0 - \Delta S_C)/[4(h - \Delta S_C)] - M_A \\ &- M_0 - \int_0^{x_0} \sigma(x_0 - x)\,dx]/(x_0 + a + b/2) \end{aligned}$ |

**Remarks:** $\Delta S_C$ is the settlement of the rock block C at $C'$, m; $T_B$ is the lateral horizontal force of the rock block B at $B'$, kN; $T_C$ is the lateral horizontal force of the rock block C at $C'$, kN; $N_B$ and $N_C$ are the shear stresses of the rock block B and C, kN; $\sigma$ is the supporting force for the roof from the coal mass in the plastic zone, MPa; $q$ is the weight from the roof and the average load of its weak overlying strata, kN m$^{-1}$; $q_0$ is the average load of the immediate roof, kN m$^{-1}$; $M_A$ and $M_B$ are residual moments of the rock beam B at $A'$ and $B'$, kN m; $M_0$ is the bending moment of the immediate roof to the roof, kN m; $K_G$ is the support coefficient of the gangues in the goaf, kN m$^{-1}$; $F_G$ is the support stress of the goaf to the roof, m; $P$ is the support resistance, kN; $x_0$ is the width of the limit equilibrium zone of the lateral coal mass, m; $a$ is the width of the roadway retained, m; $b$ is the width of the temporary support equipment, m.

where *GSI* is the geological strength index; The parameter *D* is called 'disturbance factor', which depends upon the degree of disturbance the rock mass has been subjected to by blast damage and stress relaxation. It varies from 0 for undisturbed *in situ* rock masses to 1 for very disturbed rock masses. Here, *D* is treated as 0; $m_i$ is a material constant for the intact rock. The software RocData provides the *GSI* empirical parameters and empirical values of $m_i$ of the rock mass with different lithology. The $m_b$, *s*, and *a* of different strata were calculated by the RocData.

### 4.1.2. The constitutive model of the caving zone

According to formula (3.1) and the type of the immediate roof, the height of the caving zone was calculated as 6.5 m. Then, the caving zone dimension was obtained by combining the roof-cutting parameters and the caving angle of the gob.

To use the finite-element software to simulate the compaction characteristics of the broken roof in the caving zone, a compaction theory, proposed by Salamon [27], on broken rock mass in the caving zone is applied in numerical calculation. The stress–strain relation of the rock mass was given, as follows:

$$\sigma_v = \frac{E_0 \varepsilon}{(1 - \varepsilon/\varepsilon_m)}, \tag{4.5}$$

where $\sigma_v$ is the vertical stress in the gob, MPa; $E_0$ is the initial tangent modulus of the rock in the caving zone, MPa; $\varepsilon$ is the current vertical strain; $\varepsilon_m$ is the maximum vertical strain.

$\varepsilon_m$ could be obtained by formula (4.6), as follows [21]:

$$\varepsilon_m = \frac{(b-1)}{b}, \tag{4.6}$$

where *b* is the overall expansion coefficient of the rock mass in the caving zone.

*b* could be calculated by the formula (4.7) [28]

$$b = 1 + 0.01(c_1 h + c_2), \tag{4.7}$$

where *h* is the mining height, m; $c_1$ and $c_2$ are parameters related to the roof lithology, as shown in table 1.

$E_0$ could be calculated by the formula (4.8) [29]

$$E_0 = \frac{10.39 \sigma_t^{1.042}}{b^{7.7}}, \tag{4.8}$$

where $\sigma_t$ is the uniaxial compressive strength, and *b* is the comprehensive expansion parameter of the rock mass in the caving zone.

Through formulae (4.6) and (4.7), the maximum vertical strain $\varepsilon_m$ and the comprehensive expansion coefficient *b* are 0.35 and 1.54, respectively. Then the stress–strain relation could be obtained by substituting $\varepsilon_m$ and *b* into formula (4.5). The double-yield model agreed with the stress–strain relation [30].

### 4.1.3. Calculation of the bolt and anchor cable parameters

The finite-element software was applied to simulate the bolt and anchor cable support. The beam element could not only conveniently define the anchorage lengths of the bolt and anchor cable, the cohesive strength and stiffness of the resin roll, but also could bear the tensile and shear effects of surrounding rocks [31]. Therefore, the beam element was used in the simulation. The cohesive strength and stiffness of the resin roll are essential parameters in anchor cable and bolt support. Bai *et al.* [32] have given the empirical formula of the cohesive strength $K_{bond}$, as follows:

$$K_{bond} \cong \frac{2\pi G}{10 \ln(1 + 2t/D)}, \tag{4.9}$$

where *G* is the shear modulus of the resin roll, and here is 2.25 GPa; *D* is the diameter of the bolt and anchor cable; *t* is the thickness of the resin roll.

Empirical values were provided for the numerical calculation of the cohesive strength $S_{bond}$ of the resin roll, 400 kN m$^{-1}$ [33]. The W-type steel strip and metal net can be equivalent to the structural beam element in the numerical calculation [34]. Table 3 shows the specific mechanical parameters of the bolt, anchor cable, W-shaped steel strip and metal net.

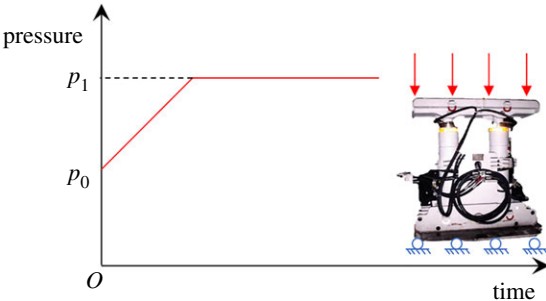

**Figure 7.** Working conditions of the temporary support equipment.

**Table 3.** Mechanical properties of the supporting materials.

| contact attributes | values |
|---|---|
| bolt/anchor cable | |
| elastic modulus (GPa) | 210 |
| yield strength (kN) | 250/470 |
| pre-tightening force (kN) | 100/150 |
| stiffness of the resin roll (N m$^{-1}$ m$^{-1}$) | $2 \times 10^9$ |
| cohesive force of the resin roll (N m$^{-1}$) | $4 \times 10^5$ |
| W-type steel strip and metal net | |
| elastic modulus (GPa) | 210 |
| tensile strength (MPa) | 500 |
| compressive strength (MPa) | 500 |
| normal stiffness of the interface (GPa m$^{-1}$) | 10 |
| shear stiffness of the interface (GPa m$^{-1}$) | 10 |

#### 4.1.4. Working conditions simulation for the temporary support equipment

The oil cylinder of the unit support and the portal support are the hydraulic components for supporting. Many engineering experiences indicate that the working resistance has two stages: the stage of increasing resistance and the stage of constant resistance during the supporting process of the oil cylinder. The special operation process of the support is the same with the ideal elastic material, so the working conditions of the support could be simulated by the constitutive model of the ideal elastic material, as shown in figure 7. Meanwhile, the thermal expansion characteristics and the initial temperature of the materials could be set to simulate the initial support force of the temporary support equipment.

#### 4.1.5. The model establishment

Based on 8201 working face in Tashan Mine in Datong Coal Mine Group Co., the numerical model was established. The length of the 8201 working face is 180 m. The continuous advancing length is 1000 m. The 5# coal seam is undermining with a mining thickness of 3.5 m and a dip of 1–4°. In figure 8, a two-dimensional model was built with a length of 290 m and a height of 68.7 m. The grid size of the model is between 0.2 and 1.0 m [35]. The generalized Hoek–Brown criterion was used for the rock mass, and the double-yield model was used in the caving zone. The bolt and anchor cable were applied in the roadway. At the same time, the slap, unit support or portal support was also used. The right and left side of the model restrained the horizontal displacement, and the bottom restrained the vertical displacement. About 10 MPa uniform vertical loads were applied on the top of the model to replace the overlying strata weight above 400 m [33], and the ground stress was also applied.

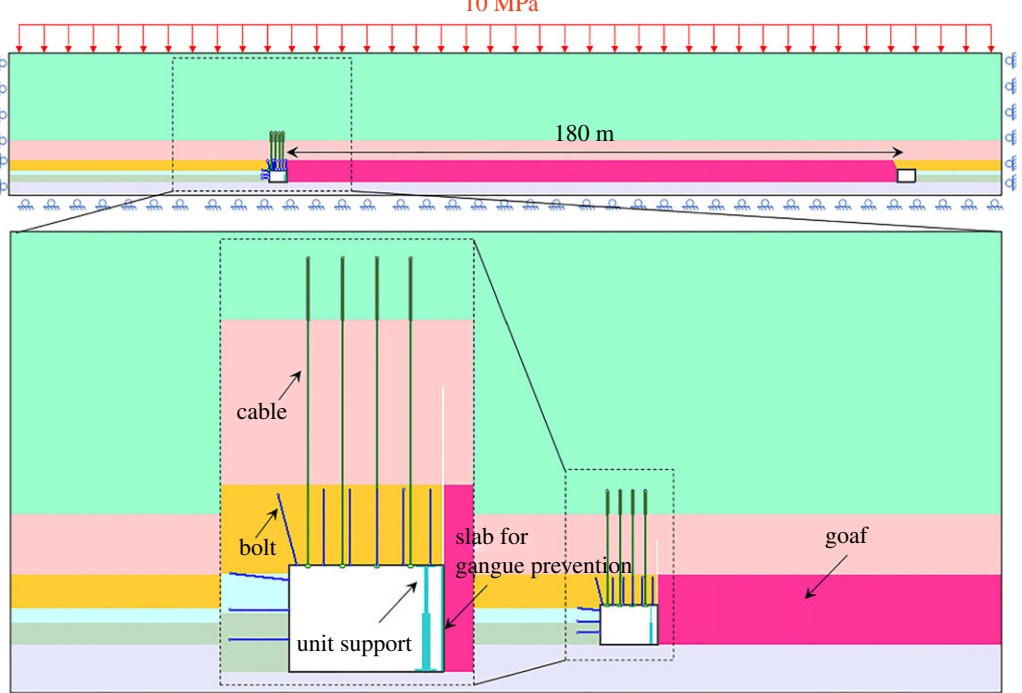

**Figure 8.** The numerical model.

## 4.2. The critical parameter optimization of the roof-cutting technology

### 4.2.1. Roof-cutting height

It can be found from figure 9, after the roof collapses, vertical stress status of the roadway surrounding rock were extracted under different roof-cutting heights, and the anchor cable and bolt tensile stresses and the resistance of the unit support were monitored.

Figure 9 indicates that:

(i) When the roof-cutting heights were 4 m, 6 m and 8 m, the axial tensions of the bolt were 126, 91 and 73 kN, respectively; the axial tensions of the anchor cable were 156, 187 and 285 kN, respectively; and the working resistances of the unit support were 44.3, 41.1 and 35.2 MPa, respectively. After the roof collapses, the roof lost constraint forces on one side of the roadway, and the force was replaced by the suspension action of the anchor cable. Therefore, as the roof-cutting height increased, the tension of the anchor cable also increased. At this point, the strata were compacted under the effect of anchor cables, so the tension of the bolt decreased. As the roof-cutting height increased, the roof thickness required to break decreases, so the working resistance for the unit support gradually reduced.

(ii) When the roof-cutting heights were 4, 6 and 8 m, the distances between positions of the stress peak in coal wall and the roadway ribs were 14.5, 18.2 and 20.5 m, respectively, and the stresses were 24.6, 23.6 and 21.2 MPa, respectively. With the increasing of the roof-cutting heights, the stress concentration peak became smaller. The longer the distance between the stress concentration position to the ribs, the more stable the roadway would be.

(iii) To further analyse the roadway surrounding rock failure conditions under different roof-cutting heights, various intensity factors of the roadway surrounding rock were given, as shown in figure 10. When the intensity factor was more than 1.0, the surrounding rock was damaged. The analysis indicated that the higher the roof-cutting height, the worse the surrounding rock failure.

The limited value of the working resistance of the unit support is 45 MPa, and the working resistance of 40 MPa was chosen for safety in this study. Meanwhile, the tension of the anchor cable was 250 kN. Based on the above requirements, the cutting height was determined as 6 m.

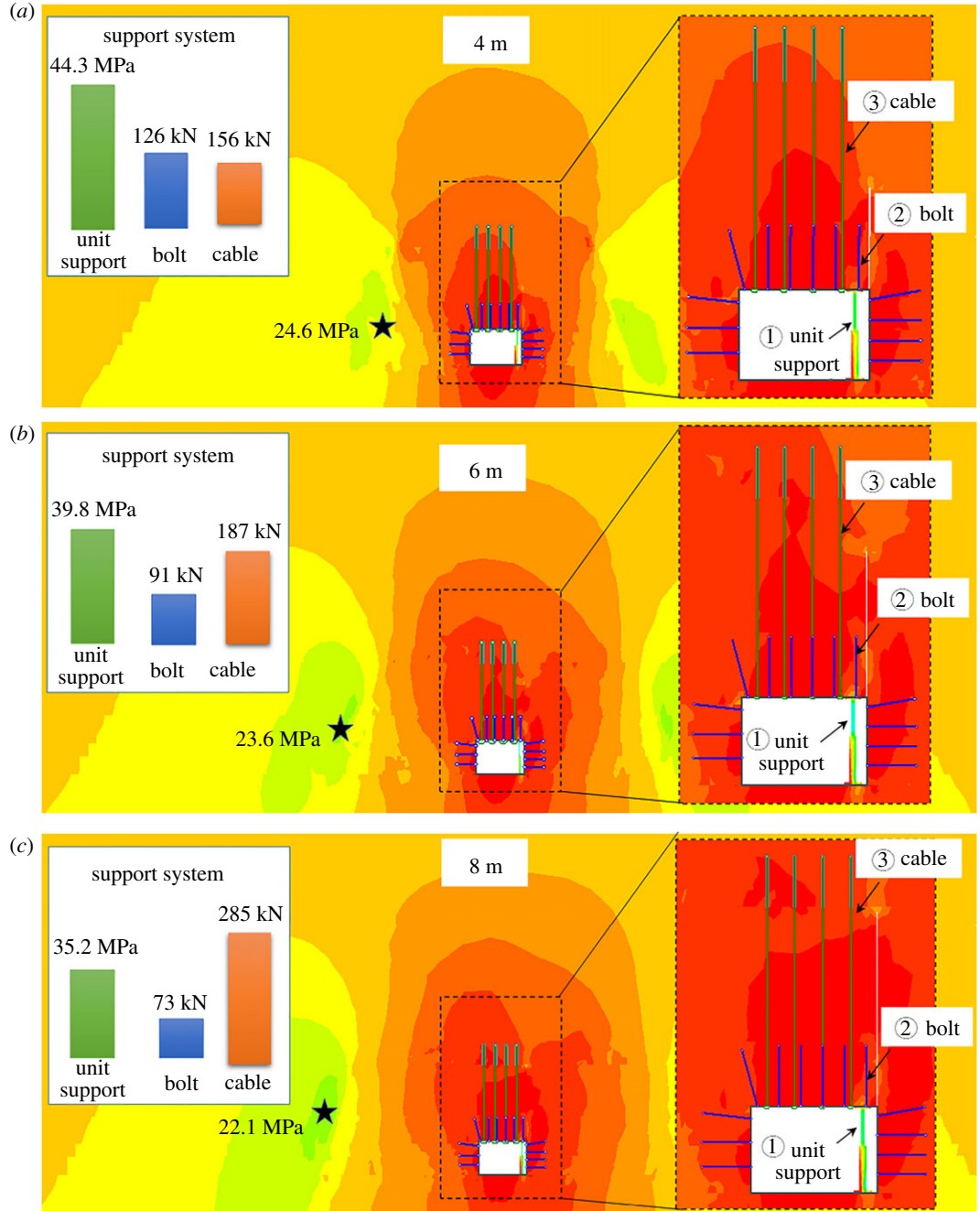

**Figure 9.** Stress states of the roadway surrounding rock. (*a*) 4 m; (*b*) 6 m; (*c*) 8 m.

### 4.2.2. Type of the temporary support equipment

The temporary support equipment is mainly used to provide specific support for the broken roof and assist in retaining the roadway. The main equipment includes the single prop, unit support and portal support. The single prop is low-cost and easy to move, but its working resistance is relatively small, and the support effects are weak. The unit support and portal support could provide more extensive support. The former could provide support for a point while the latter could provide support for the whole roadway.

Due to the high support intensity of the unit support and portal support, their support effects were mainly compared. In figure 11, the roof-cutting height was 6 m in the numerical simulation. It can be seen from figure 12, the surrounding rock vertical stress at 1 m away from the roadway roof, the lateral compressive stress of the slab, and the lateral displacement of the caving gangues were extracted.

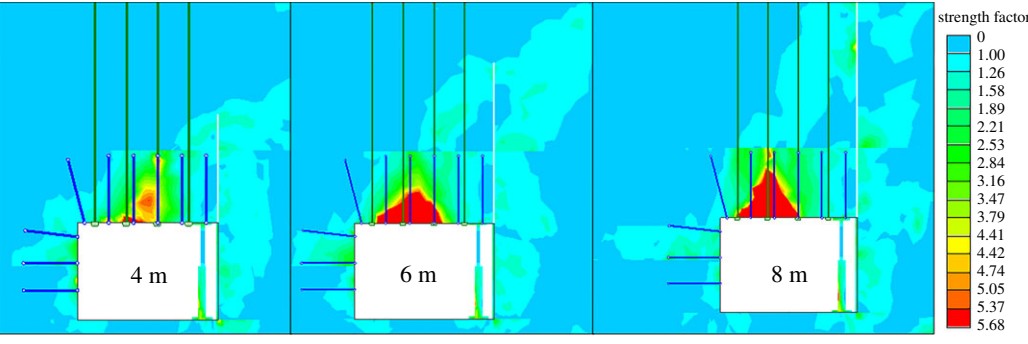

**Figure 10.** Failure zones of the roadway surrounding rocks.

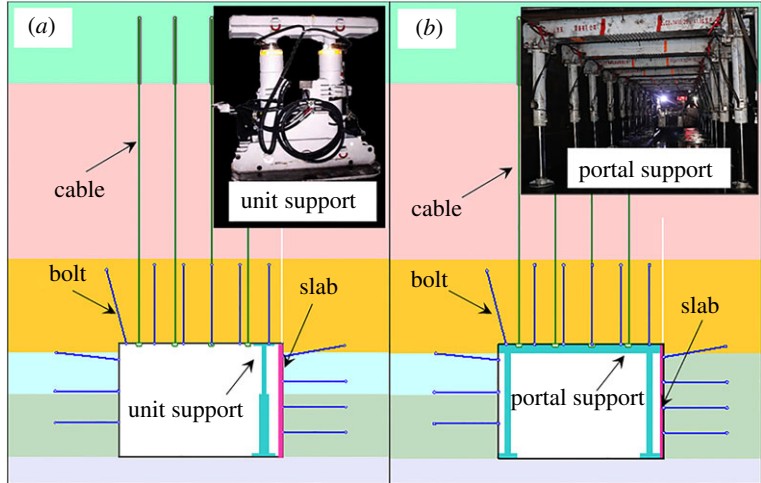

**Figure 11.** The temporary support equipment.

From figure 12a, the vertical stress of the surrounding rock with 1.0 m distance to the roadway roof was extracted. The comparative analyses indicated that: (i) the support intensity of the portal support was higher than that of the unit support; (ii) the unit support and portal support played a functional role in supporting the roof at the gob side, so the vertical stress increased sharply; and (iii) the shielding effect of the portal support was better than that of the unit support near the side of the solid coal.

The lateral extrusion stresses to the slab were extracted and summarized in figure 12b. The comparative analyses indicated that the lateral stress was significantly smaller with the portal support. The lateral extrusion stress peaks corresponding to the unit support and that to portal support were 6.2 and 4.6 MPa, respectively. Compared with the unit support, the portal support had higher working resistance and could share more significant roof pressure, thereby reducing the vertical extrusion stress from the roof to the gangues in the gob and further lowering the lateral stresses due to gangue extrusion. Figure 12c illustrates the lateral displacements of the broken roof. The comparative analysis indicated that the lateral displacement was smaller with that of the portal support. The reason was the same as the lateral extrusion stress to the slab from a broken roof.

To further compare and analyse the failure conditions of the roadway surrounding rocks with different temporary support equipment, the failure areas were given under supporting by the unit support and portal support in figure 13.

Figure 13 shows that compared with the unit support, the failure height of the roadway roof was significantly smaller supported by the portal support, and the failure area on the left side of the roadway was slightly smaller. The failure area was more severe in the middle of the roadway roof supported by unit supports.

By analysing the surrounding rock vertical stress, the lateral extrusion stress to the slab, the lateral displacement of the broken roof and failure areas of the surrounding rock, the supporting effects of the portal support were superior.

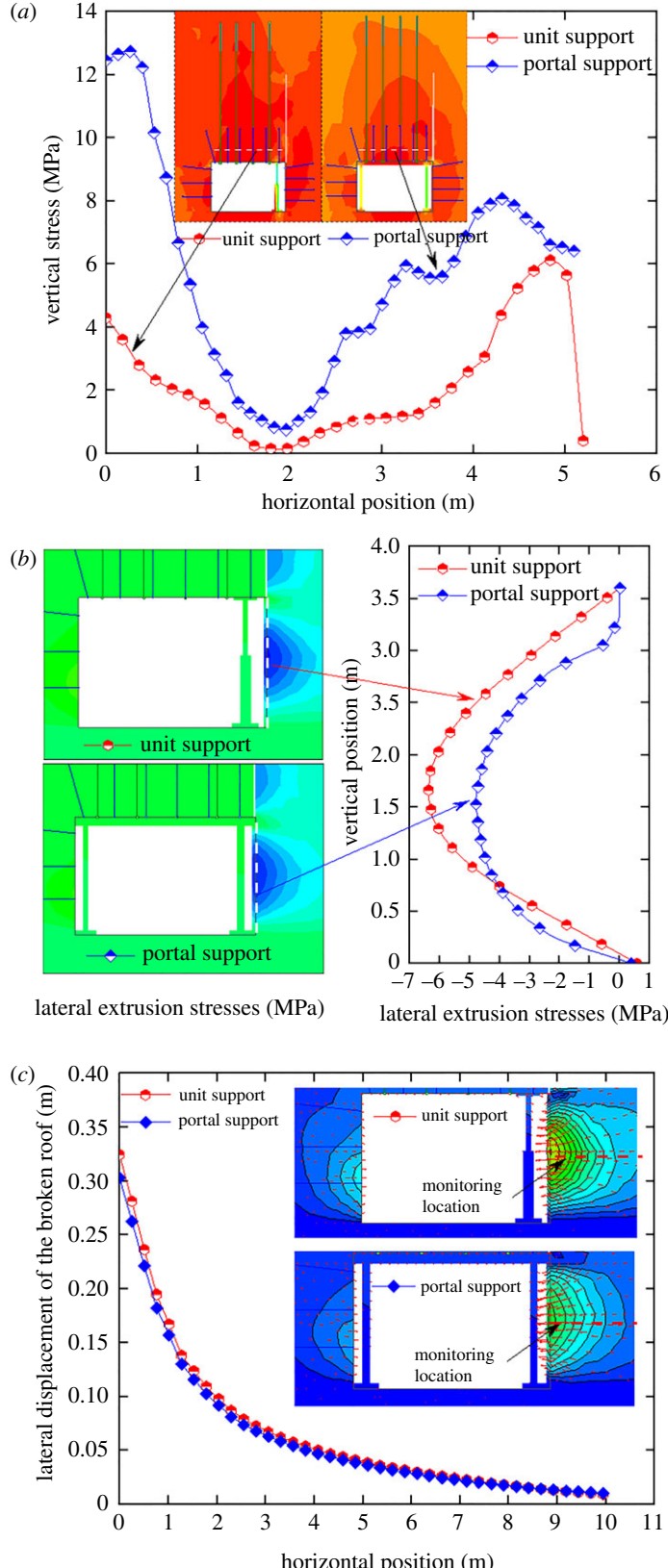

**Figure 12.** Support effects comparison of the unit support and portal support. (*a*) The vertical stress; (*b*) the lateral compressive stress to the slab; (*c*) the lateral displacement of the gangues.

### 4.2.3. The working resistance of portal support

With the advantages of higher support strength and better effects on roadway failure control, the portal support was preferred for temporary support. The resistance was an important parameter. The support

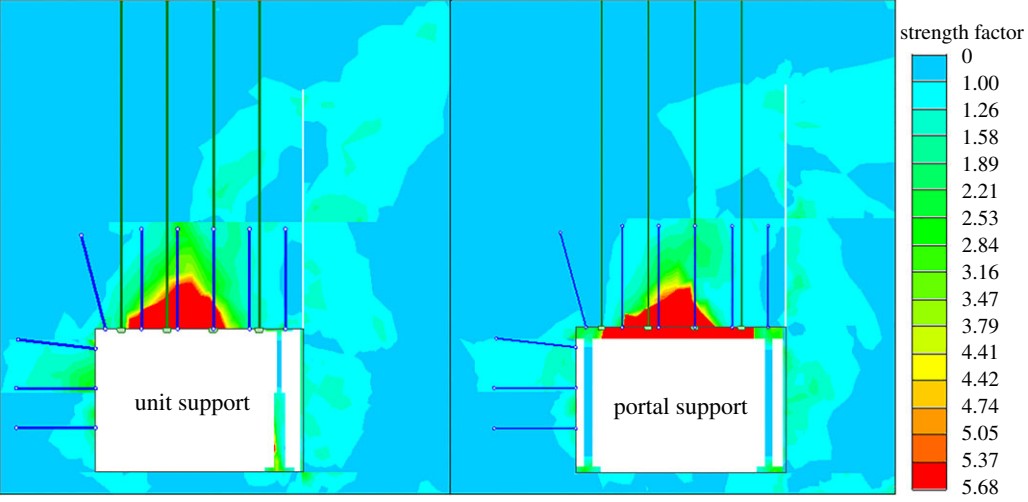

**Figure 13.** Roadway surrounding rock failure areas.

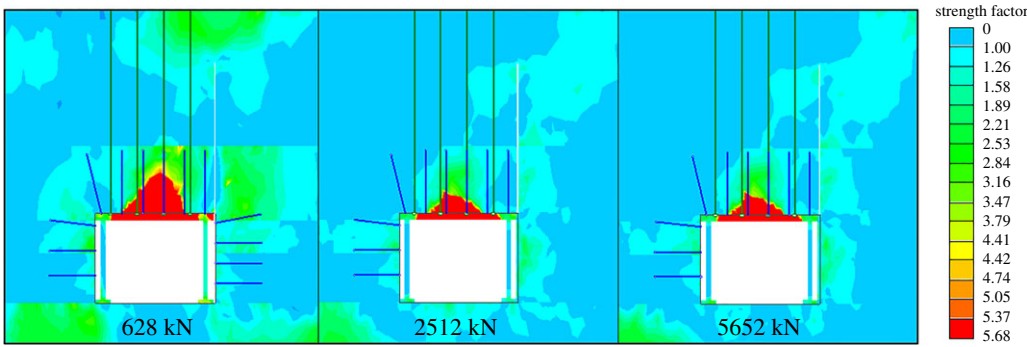

**Figure 14.** Roadway surrounding rock failure states under different working resistances.

strength was directly related to the diameter of the hydraulic cylinder. The support resistance could be obtained by formula (4.10)

$$F = \frac{\pi}{2} D^2 P_0. \tag{4.10}$$

In general, the diameter of the portal support's oil cylinder was between 100 and 300 mm and the working pressure $P_0$ was 40 MPa. When the diameters were 100, 200 and 300 mm, the corresponding supporting forces were 628, 2512 and 5652 kN, respectively. From figure 14, the failure areas of the roadway surrounding rocks were given under three working resistance by the numerical simulation method. When the support was 628 kN, the failure area was relatively large. When the support was 2512 kN, the surrounding rock failure was obviously reduced. Therefore, the diameter of the portal support's oil cylinder should be above 200 mm.

### 4.2.4. Parameters of the bolt and anchor cable

As the active support, the prestress bolt support could control dilatancy and deformation in bolted surrounding rocks, such as the roof separation, sliding, fracture opening and new crack expansion. They could keep the surrounding rocks under pressure, which could restrain the bending and deformation, tensile and shear failure, maintain the integrity of the surrounding rock and decrease the strength of the surrounding rock [36–38]. The pre-tightening force, density and length of the bolt and anchor cable need to be optimized at the design time. As shown in figure 15, the numerical simulation software FLAC was used to simulate and optimize these parameters according to the prestress theory.

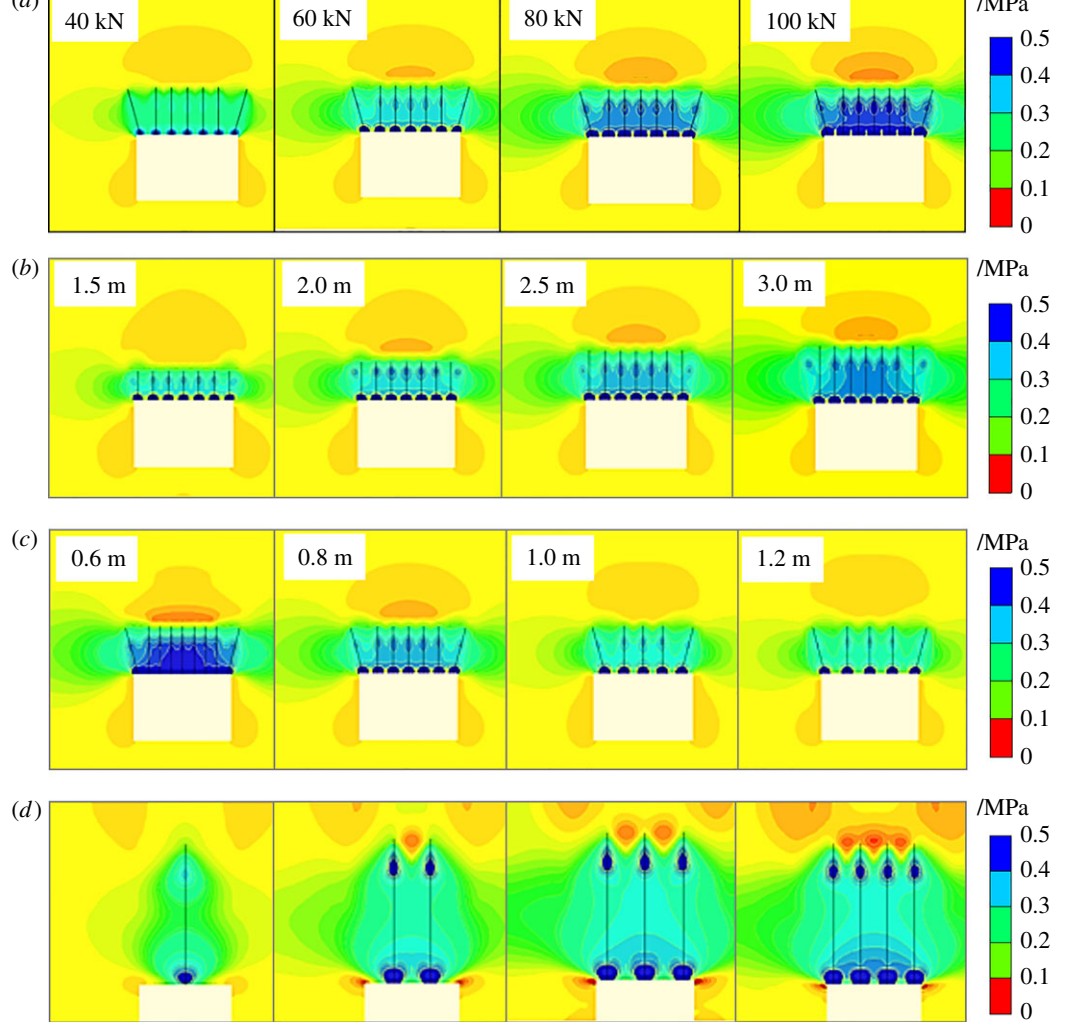

**Figure 15.** Parameters optimization of the bolt and anchor cable. (*a*) The pre-tightening force of the bolt; (*b*) the length of the bolt; (*c*) the density of the bolt; (*d*) the density of the anchor cable.

It can be found that in figure 15, the continuous compressive stress areas were formed around the roadway surrounding rock with 80 kN pre-tightening force of the bolt, 0.8 m bolt spacing, 2.0 m length of the bolt and four anchor cables. These parameters were selected as the support parameters for the roadway.

# 5. Engineering application

## 5.1. The industrial experiment

In order to adapt the underground roof-cutting process, the machine of roof cutting with chain arm was operated on the ground, which can be obtained in figure 16*a*. The skilled ground operation was conducive to master the cutting technology and provide a basis for underground engineering applications. As shown in figure 16*b*, the machine was applied in the underground roadway to conduct a roof-cutting experiment. The accumulated cutting length was 150 m, and the average daily cutting progress was 3.5 m.

## 5.2. The surrounding rock damage due to roof cutting

The kerf was formed by the machine of roof cutting with a chain arm, whose influence on the mechanical properties of roof surrounding rocks was different from that of the conventional energizing blasting roof cutting. To study the influence of roof-cutting damage of the roof surrounding rocks, the uniaxial compressive strength *in situ* test system was used to compare the uniaxial compressive strength of part with the roof cutting and the part without being cut. To ensure the accuracy of the mechanical

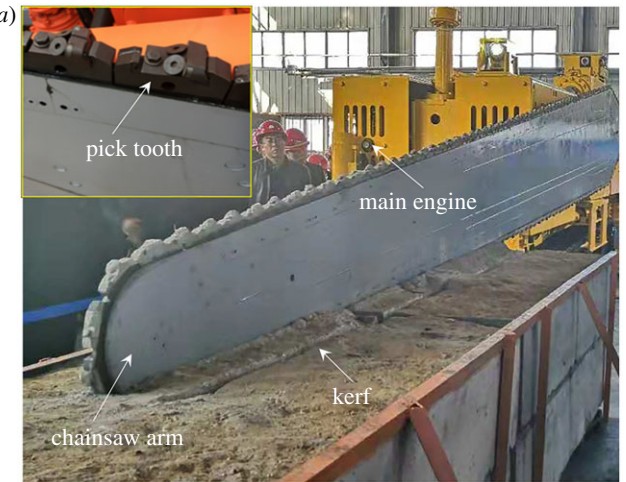

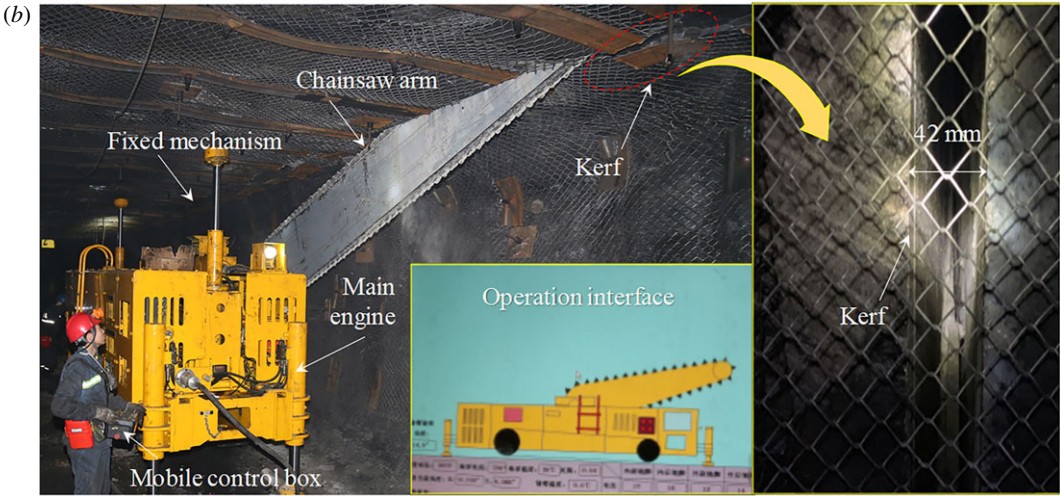

**Figure 16.** The chain arm sawing roof experiment. (*a*) Ground experiment, (*b*) underground experiment.

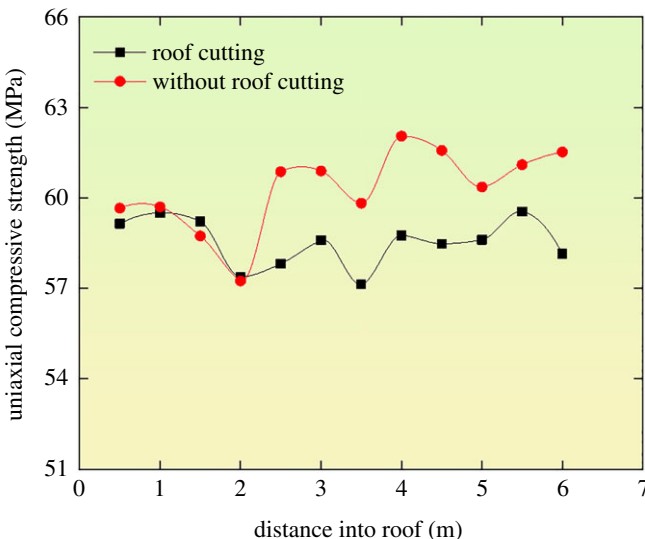

**Figure 17.** Change law of the uniaxial compressive strength of surrounding rocks.

properties change, the monitoring points were arranged at the end of the kerf formed by the chain arm and is 20 mm away from the kerf. Figure 17 shows the comparison results.

Figure 17 indicates that (i) the uniaxial compressive strength of the roof surrounding rocks with and that without roof cutting rarely changed, which was only around 5%, so the damage to surrounding rock

by roof cutting with chain arm is small, and the damage range to the roof is about 82 mm; (ii) in the range of bolt and anchor range of 0–2.5 m, the uniaxial compressive strength damage was less within the range of 0–0.6 MPa. It increased significantly beyond the bolt and anchor range of 2.5–6.0 m, and the highest was 3.37 MPa.

# 6. Conclusion

(i) Based on theoretical analysis, the working resistance of the temporary support equipment in the technology of roof cutting with chain arm to retain roadway was given.

(ii) The numerical calculation was used to optimize the roof-cutting height, the type of the temporary support equipment, working resistance of the portal support and the support parameters of the bolt and anchor cables. It was obtained that as the roof-cutting height increased, the tension of the roadway anchor cable increased, and the working resistance and the tension of the bolt decreased.

(iii) It was observed that no apparent surrounding rock damage changed slightly before and after roof cutting, and the variation fluctuation range of the uniaxial compressive strength was only around 5%. Therefore, the effect of cutting the roof and retaining the roadway is excellent to have proved its promising results.

Data accessibility. Data available from the Dryad Digital Repository: https://doi.org/10.5061/dryad.pc866t1k4 [39].
Authors' contributions. B.Y. participated in the design of the study; Y.T. drafted the manuscript; R.G. carried out the statistical analyses; Q.Y. collected field data; Z.L. conceived of the study and designed the study; H.X. coordinated the study and helped draft the manuscript. All authors gave final approval for publication.
Competing interests. We declare we have no competing interests.
Funding. This work was funded by the State Key Research Development Program of China (grant no. 2018YFC0604500); Talents Project of Liaoning Revitalization (grant no. XLYC201807219).
Acknowledgements. This work was funded by the State Key Research Development Program of China (2018YFC0604500); Talents Project of Liaoning Revitalization (XLYC201807219). The authors gratefully acknowledge the financial support from the organization.
Permission to carry out fieldwork. We obtained the appropriate permissions and licences to conduct the fieldwork detailed in our study.

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
