## [Reviewer comments · Royal Society Open Science]

Review History

RSOS-191913.R0 (Original submission)

Review form: Reviewer 1

Is the manuscript scientifically sound in its present form?

Yes

Are the interpretations and conclusions justified by the results?

Yes

Is the language acceptable?

No

Do you have any ethical concerns with this paper?

No

Have you any concerns about statistical analyses in this paper?

No

Recommendation?

Major revision is needed (please make suggestions in comments)

Comments to the Author(s)

The manuscript requires some solid editing by a native English speaker. Articles are missing, familiar language inappropriate for scientific journals must be corrected.

Some mention of the findings should be found in the abstract. Simply saying something is very good is general, ambiguous and of little value to the reader.

The authors have relied almost exclusively on Chinese literature. I am certain there are more scholars globally.

Review form: Reviewer 2

Is the manuscript scientifically sound in its present form?

Yes

Are the interpretations and conclusions justified by the results?

Yes

Is the language acceptable?

No

Do you have any ethical concerns with this paper?

No

Have you any concerns about statistical analyses in this paper?

No

Recommendation?

Major revision is needed (please make suggestions in comments)

Comments to the Author(s)

The study conducted is an asset to the underground coal mining industry and hence suggested for a revision. Especially in English.

I did make some comments and highlighted a few places as an example in an attached annotated copy of the paper (Appendix A).

Conclusions need to be punchy with outcomes and the achievements that are beneficial to the coal mining community.

Decision letter (RSOS-191913.R0)

17-Jan-2020

Dear Dr Tai,

The editors assigned to your paper ("The Sustainable Development of Coal Mines by New Cutting Roof Technology") have now received comments from reviewers. We would like you to revise your paper in accordance with the referee and Associate Editor suggestions which can be found below (not including confidential reports to the Editor). Please note this decision does not guarantee eventual acceptance.

Please submit a copy of your revised paper before 09-Feb-2020. Please note that the revision deadline will expire at 00.00am on this date. If we do not hear from you within this time then it will be assumed that the paper has been withdrawn. In exceptional circumstances, extensions may be possible if agreed with the Editorial Office in advance. We do not allow multiple rounds of revision so we urge you to make every effort to fully address all of the comments at this stage. If deemed necessary by the Editors, your manuscript will be sent back to one or more of the original reviewers for assessment. If the original reviewers are not available, we may invite new reviewers.

- Data accessibility

If you wish to submit your supporting data or code to Dryad (<http://datadryad.org/>), or modify your current submission to dryad, please use the following link:
<http://datadryad.org/submit?journalID=RSOS&manu=RSOS-191913>

- Competing interests

- Authors' contributions

- Acknowledgements

- Funding statement

on behalf of Prof R. Kerry Rowe (Subject Editor)
openscience@royalsociety.org

Associate Editor's comments:

Thank you for your submission to Royal Society Open Science. Whilst the reviewers are supportive of publication, they recommend major revisions. Please ensure to provide a thorough response to all comments made. In your revised version, please ensure to provide:

- A clean copy of your revised manuscript;
- A tracked edits version highlighting the revisions you have made;

As you have been requested to edit the written English, you must provide proof that you have done so: acceptable proof includes a certificate of language-editing from a language editing service or a signed letter from a native speaker of English. If you do not provide this proof, your manuscript may be returned to you.

For information about language editing services endorsed by the Royal Society, please follow the link below:

<https://royalsociety.org/journals/authors/language-polishing/>

Comments to Author:

Reviewers' Comments to Author:

Reviewer: 1

Comments to the Author(s)

The manuscript requires some solid editing by a native English speaker. Articles are missing, familiar language inappropriate for scientific journals must be corrected.

Some mention of the findings should be found in the abstract. Simply saying something is very good is general, ambiguous and of little value to the reader.

The authors have relied almost exclusively on Chinese literature. I am certain there are more scholars globally.

Reviewer: 2

Comments to the Author(s)

The study conducted is an asset to the underground coal mining industry and hence suggested for a revision. Especially in English.

I did make some comments and highlighted a few places as an example in an attached annotated copy of the paper

Conclusions need to be punchy with outcomes and the achievements that are beneficial to the coal mining community.

Author's Response to Decision Letter for (RSOS-191913.R0)

See Appendix B.

RSOS-191913.R1 (Revision)

Review form: Reviewer 2

Is the manuscript scientifically sound in its present form?

Yes

Are the interpretations and conclusions justified by the results?

Yes

Is the language acceptable?

Yes

Do you have any ethical concerns with this paper?

No

Have you any concerns about statistical analyses in this paper?

No

Recommendation?

Accept as is

Comments to the Author(s)

The paper is good and believes to be an asset to the industry and hence suggesting for publication. I still believe there is room for improvement when "English writing" is concerned. Please consider that in future publications or even in the final version of this paper.

Decision letter (RSOS-191913.R1)

14-Apr-2020

Dear Dr Tai:

On behalf of the Editors, I am pleased to inform you that your Manuscript RSOS-191913.R1 entitled "The Sustainable Development of Coal Mines by New Cutting Roof Technology" has been accepted for publication in Royal Society Open Science subject to minor revision in accordance with the referee suggestions. Please find the referees' comments at the end of this email.

The reviewers and Subject Editor have recommended publication, but also suggest some minor revisions to your manuscript. Therefore, I invite you to respond to the comments and revise your manuscript.

- Ethics statement

- Data accessibility

<http://datadryad.org/submit?journalID=RSOS&manu=RSOS-191913.R1>

- Competing interests

- Authors' contributions

AB carried out the molecular lab work, participated in data analysis, carried out sequence alignments, participated in the design of the study and drafted the manuscript; CD carried out the statistical analyses; EF collected field data; GH conceived of the study, designed the study,

coordinated the study and helped draft the manuscript. All authors gave final approval for publication.

- Acknowledgements

- Funding statement

Because the schedule for publication is very tight, it is a condition of publication that you submit the revised version of your manuscript before 23-Apr-2020. Please note that the revision deadline will expire at 00.00am on this date. If you do not think you will be able to meet this date please let me know immediately.

on behalf of Mr Andrew Dunn (Associate Editor) and R. Kerry Rowe (Subject Editor)
openscience@royalsociety.org

Associate Editor Comments to Author:

Please accept our apologies for the delay in your paper being reviewed. We've now received a report from a referee who recommends acceptance, and the Editors agree from a scientific perspective; however, it is important that you also seek professional English language editing support (<https://royalsociety.org/journals/authors/benefits/language-editing/>) before resubmitting - indeed, acceptance is considered contingent on your demonstrating that further linguistic advice has been received. If you require a short extension to your deadline to allow you to receive this advice, please contact openscience@royalsociety.org for support.

Reviewer comments to Author:

Reviewer: 2

Comments to the Author(s)

The paper is good and believes to be an asset to the industry and hence suggesting for publication. I still believe there is room for improvement when "English writing" is concerned. Please consider that in future publications or even in the final version of this paper.

Author's Response to Decision Letter for (RSOS-191913.R1)

See Appendix C.

Decision letter (RSOS-191913.R2)

04-May-2020

Dear Dr Tai,

It is a pleasure to accept your manuscript entitled "The Sustainable Development of Coal Mines by New Cutting Roof Technology" in its current form for publication in Royal Society Open Science.

on behalf of Mr Andrew Dunn (Associate Editor) and R. Kerry Rowe (Subject Editor)
openscience@royalsociety.org

Appendix A**ROYAL SOCIETY
OPEN SCIENCE****The Sustainable Development of Coal Mines by New Cutting
Roof Technology**

Journal:	Royal Society Open Science
Manuscript ID	RSOS-191913
Article Type:	Research
Date Submitted by the Author:	31-Oct-2019
Complete List of Authors:	Yu, Bin; Datong Coal Mine Group Co Ltd Tai, Yang; China University of Mining and Technology; china university of mining and technology Gao, Rui; Taiyuan University of Technology Yao, Qiangling; China University of Mining and Technology Li, Zhao; Datong Coal Mine Group Co Ltd Xia, Hongchun; Dalian University, College of Civil and Architectural Engineering
Subject:	Engineering geology < ENGINEERING AND TECHNOLOGY, Energy < ENGINEERING AND TECHNOLOGY
Keywords:	sustainable development, coal mine, roof cutting, chain arm, retain roadway
Subject Category:	Engineering

Author-supplied statements

Relevant information will appear here if provided.

Ethics

Does your article include research that required ethical approval or permits?:

This article does not present research with ethical considerations

Statement (if applicable):

CUST_IF_YES_ETHICS :No data available.

Data

It is a condition of publication that data, code and materials supporting your paper are made publicly available. Does your paper present new data?:

Yes

Statement (if applicable):

All original data are deposited at Figshare:
<https://figshare.com/s/c9a6404ed59ae8e66f62>.

Conflict of interest

I/We declare we have no competing interests

Statement (if applicable):

CUST_STATE_CONFLICT :No data available.

Authors' contributions

This paper has multiple authors and our individual contributions were as below

Statement (if applicable):

B.Y participated in the design of the study; Y.T. drafted the manuscript; R.G. carried out the statistical analyses; Q.Y. collected field data; Z.L. conceived of the study and designed the study; H.X. coordinated the study and helped draft the manuscript. All authors gave final approval for publication.

The Sustainable Development of Coal Mines by New Cutting Roof Technology

Bin Yu¹, Yang Tai^{2,*}, Rui Gao³, QiangLing Yao², Zhao Li¹, Hongchun Xia⁴

¹ Datong Coal Mine Group Co. Ltd., Datong, 037000, China;

² School of Mines, China University of Mining and Technology, Xuzhou 221116, China;

³ College of Mining Engineering, Taiyuan University of Technology, Shanxi 030024, China;

⁴ College of Architectural Engineering, Dalian University, Liaoning, Dalian 116622, China.

* Correspondence author: cumtcqy@163.com

Abstract: China consumes more than 3.6 billion tons of coal every year, accounting for over 50% of the global coal consumption. Therefore, the sustainable development of coal mines is a problem needed to be solved by Chinese government. During the coal resources recovery, the serious loss for coal resources is caused by the protective coal pillars between the adjacent working faces. In order to solve the problem, it was put forward that the new technology of roof cutting with chain arm to retain roadway in the paper. Firstly, the process of retaining roadway, roof-cutting parameters and the damage ranges of roadway surrounding rock induced by roof cutting with chain arm were analyzed. Then the roof-cutting height, the type of temporary support equipment, working resistance of portal support and support parameters of the bolt and anchor cables were optimized by the numerical calculation. Finally, the industrial experiment of retaining roadway by roof cutting with chain arm was carried out in a working face. It was obtained that the surrounding rock damage changed slightly before and after roof cutting, and the variation range of the uniaxial compressive strength was only around 5%. The effect of cutting roof and retaining roadway is very good.

Keywords: sustainable development, coal mine, roof cutting, chain arm, retain roadway

1 Introduction

China is the largest coal consumer in the world and its coal consumption accounts for more than 60% of primary energy consumption [1]. However, with the high strength exploitation, coal resources are quickly exhausted, especially in China's eastern provinces, such as Jiangsu, Henan and Shandong and etc. [2]. During the process of coal resources recovery, the protective coal pillars between adjacent working faces have been a major cause for low recovery rate[3]. They are used to isolate goaf for safe production for the next adjacent working faces. According to the width for the coal pillars, it can be divided into large-pillar method and small-pillar method [4, 5]. As shown in Figure 1, the large-pillar method is to arrange the roadway in in-situ stress area to avoid impacts of abutment pressure. This method is beneficial to maintain the roadway, but it seriously cause loss of coal resources [6]. In order to reduce the loss of coal pillars and improve coal recovery rate in coal mines, small-pillar method has been put forward. It is to arrange the roadway in the stress reduction area of the surrounding rock. The width of small coal pillars is generally between 3 m and 6 m [6]. This method could make the roadway in the stress reduction area where the roadway is in an easy maintenance and the loss of coal resources could be effectively avoided. Thus, the small-pillar method has become a priority for the roadway arrangement [7, 8]. However, the advancing part for the roadway should be increased, because the small coal pillars are arranged in the stress reduction area and the roadway surrounding rocks are in the plastic area [9, 10]. Besides, the small-pillar method must be carried out after the strata stability of the upper working face which could cause the problem for the mining of next working face in time. At the same time, the roadway along the gob is in the plastic failure area of the surrounding rock, as shown in Figure 2. When there are thick-hard strata above the working face, the damage of coal pillars is more serious and the coal rib spalling occurs near goaf under the dynamic pressure of the thick-hard strata [11].

Fig.1 Layout method of coal pillars

Fig.2 The roadway failure areas in the small pillar method

In order to further increase the coal recovery rate, realize sustainable development for mines and reduce the pressure for the mining of next working face in time, non-pillar mining method has been put forward. The roadway roof and the goaf roof is a whole structure in traditional mining methods, and their movements were highly correlated. The roof breaks and fills the roadway under the actions of mining pressure, as shown in Figure 3 (a). Compared with the traditional roadway treatment methods, the non-pillar method, which cuts roof to retain gob-side roadway, is to conduct drilling hole and installing explosives in the roadway roof near goaf, and then implement roof presplitting by directional energizing blasting technology. As shown in Figure 3 (b), after the working face passes, the slap to prevent gangue should be timely adopted to avoid the slip of broken roof into the roadway. Meanwhile, the presplitting roof is crushed into the goaf under the actions of surrounding rock pressure in the stope and the temporary supporting equipment. Under the influence of expansion of rock mass, the broken roof fills the goaf and provides a certain supporting force to balance and stabilize the strata in the goaf. The broken roof could basically support the overburden, control the subsidence to a certain degree and reduce the loads for coal pillar from cantilever beam structure, thereby guaranteeing the roadway stability. The collapsed roof formed one wall for the roadway, and the preserved roadway will service for the next mining of the working face.

Fig.3 The treatment methods for roadway

At present, many **researches** on non-pillar method have been carried out. He et al. [12, 13] proposed non-pillar method and developed the constant-resistance and large-deformation anchor cable for this technology. The mine pressure law of the non-pillar working face was revealed. The working resistance for temporary support equipment was given, and the theory and technology of the non-pillar method was established. Gao et al. [14, 15] studied the dynamic impact behaviors of the gangue body under different mining heights. The mechanism and control techniques for gangue rib deformations were explored in detail. A new control approach was developed to solve the instability problem of the gangue rib in thick coal seams. Sun et al.[16] studied key parameters in non-pillar method to retain goaf-side roadway for thin coal seams mining. The roof-cutting height, presplitting angle and the distance between presplitting blasting holes were optimized and determined. Guo et al. [17] gave the theoretical formulas for the roof-cutting height and presplitting angle based on theoretical calculation.

The aforementioned researches mainly focus on non-pillar mining method to retain goaf-side roadway from the aspect of theory and technology. And the energizing blasting technology is usually adopted as the roof-cutting method. It has a complicated operation process in drilling hole, installing explosive, detonating explosive and etc. It also has a low mechanization degree, and large disturbance damage to the roof. The broken roof has poor self-standing ability. Besides, the directional blasting is hard to be accurate and fracture between adjacent boreholes could not be ensured for penetration. **What's more, blasting is prohibited in mines with high gas.** Aiming at the shortcomings of the energizing blasting, a weak-disturbance, high-efficiency, accurate and continuous roof-cutting method is in urgent need. Datong Coal Mine Group Co., Ltd. has developed the technology and roof-cutting equipment with chain arm to retain roadway. The new technology could effectively address the aforementioned problems, improve the recovery rate of coal resources, prolong service life of mines and ensure their sustainable development.

The structure characteristics of the roof-cutting equipment with chain arm and the process of roof cutting were firstly introduced in the paper. Next, the process of retaining roadways, roof-cutting parameters and damage range were analyzed. Then the key technical parameters for the new technology were optimized by numerical calculation. Finally, the industrial experiment of hard roof cutting was carried out in Datong Coal Mine Group Co., Ltd.

2 The equipment of roof cutting with chain arm and technology of retaining roadway

The roof cutting with chain arm to retain roadway is to continuously and accurately cut the roof with weak disturbance and high efficiency by the chain arm assisted with laser calibration system. It could provide the basis for retaining gob-side roadway and improve the recovery rate of coal resources. The roof cutting equipment and technology of retaining roadway were mainly introduced in this chapter.

2.1 The equipment of roof cutting with chain arm

The key equipment is the machine of roof cutting with chain arm. As shown in Figure 4, its components include chain arm, crawler travel mechanism, fixed mechanism, power unit, spray system and etc. It has the following advantages: (1) no water needed; (2) low-speed rotate and not easy to produce sparks; (3) without pre-drilling and save auxiliary time; (4) smooth cutting surface, good shaping; (5) continuous cutting, high efficiency, low labor intensity and labor-saving operation; (6) use of the round polycrystalline diamond; the hardness of 10 times higher than traditional materials; usually no less than 1000 hours service life; (7) use of internal and external spray system, effective dust control; (8) effective and accurate tool setting by laser calibration system.

Fig.4 The machine of roof cutting with chain arm

2.2 The process of roof cutting with chain arm

As shown in Figure 5, roof-cutting process is as follows: (1) To start the crawler travel mechanism and place the machine at the end of the roadway; (2) To use the crawler travel mechanism to make the chain arm close to coal wall of the working face, and keep the machine body to be parallel to the roadway; (3) To use the oil cylinder of the fixed mechanism to make the crawler travel mechanism maintain a distance of 10 mm to 20 mm above the ground; (4) To start the power unit and drive the rotation of chain, and rotate the chain arm from one side to the other side at constant speed of 180°; (5) To rapidly rotate to the initial horizontal position after moving to the other side; (6) To adopt the crawler travel mechanism to make the continuous cutting machine move forward a certain distance; (7) To use laser calibration system to keep the chain arm accurately align with the above cutting crack, and to repeat the process of (2) to (6) and begin the next roof cutting.

Fig.5 The process of roof cutting with chain arm

3 The technology of roof cutting with chain arm to retain roadway

3.1 The process of retaining roadway

The technology could be divided into 3 steps, as shown in Figure 6. Step 1: The machine of roof cutting with chain arm is used to cut the roof; Step 2: after the working face pass, the roadway temporary support should be strengthened and the gangues should be prevented timely at the rear of the working face. The special unit support or portal support is mainly used as the temporary support. The slap is used to prevent broken roof into the roadway. Step 3: The broken roof in goaf is gradually compacted as the working face advances. The temporary support equipment can be removed after roof stabilization. The sprayed concrete is used for the rib formed by broken roof to prevent air leakage. Finally, the process is completed.

Fig.6 The process of retaining roadway

3.2 Key technical parameters

In the technology of roof cutting with chain arm to retain roadway, the key technical parameters are important

127 to guarantee effects of retaining roadway. They include the roof-cutting height, the type and working resistance for
 128 temporary support and support parameters of bolt and cable. This chapter mainly analyzed the roof-cutting height
 129 and working resistance of the temporary support equipment.

130 3.2.1 Roof-cutting height

131 The roof-cutting height H refers to the maximum vertical height of the kerf. In general, the height should be
 132 more than the caving zone H_0 , that is $H > H_0$, in order to guarantee that the broken roof could backfill the goaf and
 133 the ensure the stability of the strata. The scholars have obtained the regression formula for the height of the caving
 134 zone through analyzing the caving zones ranges in a large number of mines with different geological conditions in
 135 China and US, as follows [18]:

$$136 H_0 = 100h / (c_1 h + c_2) \quad (1)$$

137 where h is the mining height, m; c_1 and c_2 are parameters related with roof lithology, as shown in table 1.

138 **Table 1 Coefficients for the caving zone height**

Type of the immediate roof	Uniaxial compressive strength /MPa	Coefficients	
		c_1	c_2
Hard	>40	2.1	16
Mid-hard	20~40	4.7	19
Weak	<20	6.2	32

139 Therefore, the roof-cutting height should satisfy: $H > 100h / (c_1 h + c_2)$.

140 3.2.2 Working resistance of the temporary support equipment

141 There are three broken positions for the roof in the large-pillar method and small-pillar method, that is outside
 142 of the coal pillar, directly above the roadway and inside of the solid coal wall. The influencing factors include strata
 143 thickness, strata mechanical properties, mining depth, in-situ rock stress state, mining height and etc. Compared
 144 with the large-pillar method and small-pillar method, the broken position of the roof is at the boundary of the goaf
 145 by the technology of roof cutting with chain arm to retain roadway. He et al. [19] have given the working resistance
 146 for the temporary support equipment, as shown in Table 2.

147 **Table 2 Working resistance of the temporary support equipment [19]**

148 3.3 The disturbance ranges of roof cutting to the roadway roof

149 Compared with roof cutting by the energizing blasting, the technology of roof cutting with chain arm could
 150 reduce the damage to the roadway roof. The rock damage range of the energizing blasting is as follows:

$$151 R_s = r_b \left[\frac{\lambda P_b}{(1 - D_0) \sigma_t + p} \right]^{\frac{1}{\alpha}} \quad (2)$$

152 where r_b is the radius of the bore; λ is the coefficient of lateral pressure, $\lambda = \mu / (1 - \mu)$; μ is the dynamic Poisson's
 153 ratio of the roof rock mass; D_0 is the initial damage of the rock mass; σ_t is the tensile strength of the roof rock mass;
 154 p is the in-situ rock stress; α is the attenuation index of the explosive stress wave in the rock mass, $\alpha = 2 - \mu / (1 - \mu)$,
 155 which is related with roof lithological characters and blasting method; P_b is the peak pressure of the shock wave in
 156 the pore wall.

157 To achieve the penetration effect between boreholes and control the damage range of the surrounding rocks
 158 from the blasting, the distance between adjacent boreholes is generally between 500 mm and 1000 mm. Therefore,
 159 the range rock damage area by the traditional energizing blasting is between 500 mm and 1000 mm.

160 For the technology of roof cutting with chain arm to retain roadway, the rock damage range is usually less than
 161 0.5 times of the kerf width which is usually 42 mm, so the damage range for the rock is less than 84 mm [20, 21].
 162 The technology of roof cutting with chain arm, by contrast, has smaller damage range to the roadway roof and is
 163 more conducive to the maintenance of the roadway roof.

164 4 The optimization for key parameters of the technology of roof cutting with chain 165 arm to retain roadway

166 The key parameters include the roof-cutting height, the type and working resistance of the temporary support
167 equipment and the support parameters of bolt and anchor cable. They were optimized by the numerical simulation
168 method to ensure the effects of the roof cutting with chain arm to retain roadway.

169 4.1 The numerical model

170 4.1.1 Rock mass strength criterion at the stope

171 Due to a large number of irregular joints and cracks in the rock mass, the rock parameters obtained from
172 laboratory are usually higher than those at the stope. To simulate the strength reduction of the coal and rock mass,
173 Hoek & Bray proposed the Hoek-Brown failure criterion in 1984, which was revised as the Generalized Hoek-
174 Brown criterion [22, 23] and could be expressed as follows:

$$175 \sigma_1 = \sigma_3 + \sigma_{ci} \left(m_b \frac{\sigma_3}{\sigma_{ci}} + s \right)^a \quad (3)$$

176 Where σ_1 is the maximum principal stress under damage; σ_3 is the minimum principal stress; m_b is a reduced
177 value (for the rock mass) of the material constant m_i (for the intact rock); s and a are constants which depend upon
178 the characteristics of the rock mass; σ_{ci} is the uniaxial compressive strength (UCS) of the intact rock pieces. The
179 expressions of m_b , s and a are as follows:

$$180 m_b = m_i \exp\left(\frac{GSI - 100}{28 - 14D}\right) \quad (4)$$

$$181 s = \exp\left(\frac{GSI - 100}{9 - 3D}\right) \quad (5)$$

$$182 a = \frac{1}{2} + \frac{1}{6} \left(e^{-GSI/15} - e^{-20/3} \right) \quad (6)$$

183 Where GSI is the Geological Strength Index; The parameter D is a "disturbance factor" which depends upon
184 the degree of disturbance to which the rock mass has been subjected by blast damage and stress relaxation. It varies
185 from 0 for undisturbed in situ rock masses to 1 for very disturbed rock masses. Here is 0; m_i is a material constant
186 for the intact rock. The software RocData provides the GSI empirical parameters and empirical values of m_i of the
187 rock mass with different lithology. The m_b , s and a of different strata were calculated by RocData.

188 4.1.2 The constitutive model of the caving zone

189 According to formula (1) and the type of the immediate roof, the height of the caving zone was 6.5 m. Then
190 the caving zone was obtained by combining the roof-cutting parameters and the caving angle of the goaf.

191 To use the finite element software to simulate the compaction characteristics of broken roof in the caving zone,
192 Salamon [24] put forward the compaction theory of the broken rock mass in the caving zone, that is, the stress-strain
193 relation of the rock mass was given, as follows:

$$194 \sigma_v = E_0 \varepsilon / (1 - \varepsilon / \varepsilon_m) \quad (7)$$

195 Where σ_v is the vertical stress in the goaf, MPa; E_0 is the initial tangent modulus of the rock in the caving
196 zone, MPa; ε is the current vertical strain; ε_m is the maximum vertical strain.

197 ε_m could be obtained by formula (8), as follows [18]:

$$198 \varepsilon_m = (b-1) / b \quad (8)$$

199 where b is the comprehensive expansion coefficient of the rock mass in the caving zone.

b could be calculated by formula (9) [25]:

$$b=1+0.01(c_1h+c_2) \quad (9)$$

where h is the mining height, m; c_1 and c_2 are parameters related with the roof lithology, as shown in Table 1.

E_0 could be calculated by formula (10) [26]:

$$E_0=10.39\sigma_t^{1.042}/b^{7.7} \quad (10)$$

where σ_t is the uniaxial compressive strength, and b is the comprehensive expansion parameter of the rock mass in the caving zone.

Through formulas (8) and (9), the maximum vertical strain ε_m and the comprehensive expansion coefficient b are 0.35 and 1.54, respectively. Then the stress-strain relation could be obtained by substituting ε_m and b into formula (7). The Double-Yield model agreed with the stress-strain relation [27].

4.1.3 Calculation of the bolt and anchor cable parameters

The finite element software was applied to simulate the bolt and anchor cable support. The beam element could not only conveniently define the anchorage lengths of the bolt and anchor cable, and the cohesive strength and stiffness of the resin roll, but also could bear the tensile and shear effects of surrounding rocks [28], so the beam element was used in the simulation. The cohesive strength and stiffness of the resin roll are essential parameters in anchor cable and bolt support. Bai et al.[29] have given the empirical formula of the cohesive strength K_{bond} , as follows:

$$K_{bond} \cong \frac{2\pi G}{10\ln(1+2t/D)} \quad (11)$$

Where G is the shear modulus of the resin roll, and here is 2.25 GPa; D is the diameter of the bolt and anchor cable; t is the thickness of the resin roll.

It was provided that empirical values for the numerical calculation of the cohesive strength S_{bond} of the resin roll, and the special parameter is 400 kN/m [30]. The W-type steel strip and metal net can be equivalent to structural beam element in the numerical calculation [31]. Table 3 shows the specific mechanical parameters of the bolt, anchor cable, W-type steel strip and metal net.

Table 3 Mechanical properties of the supporting materials

Contact attributes	Values
Bolt/anchor cable	
Elastic modulus/GPa	210
Yield strength/kN	250/470
Pre-tightening force/kN	100/150
Stiffness of the resin roll /N/m/m	2e9
Cohesive force of the resin roll /N/m	4e5
W-type steel strip and metal net	
Elastic modulus/GPa	210
Tensile strength/MPa	500
Compressive strength/MPa	500
Normal stiffness of the interface/GPa/m	10
Shear stiffness of the interface/GPa/m	10

4.1.4 Working conditions simulation for the temporary support equipment

The oil cylinder of unit support and the portal support are the hydraulic components for supporting. Many engineering experiences indicates that the working resistance has two stages: the stage of increasing resistance and the stage of constant resistance during the supporting process of the oil cylinder. The special operation process of the support is the same with the ideal elastic material, so the working conditions of the support could be simulated by the constitutive model of the ideal elastic material. Meanwhile, the thermal expansion characteristics and the initial temperature of the materials could be set to simulate the initial support force of the temporary support equipment.

Fig.7 Working conditions of the temporary support equipment**4.1.5 The model establishment**

Based on 8201 working face in Dadougou Mine in Datong Coal Mine Group Co. as the engineering background, the numerical model was established. The length of 8201 working face is 180 m. The continuous advancing length is 1000 m. The 5# coal seam is under mining with the mining thickness of 3.5 m and dip of $1\sim 4^\circ$. As shown in Figure 8, a two-dimensional model was built with the length of 290 m and the height of 68.7 m. The grid size of the model is between 0.2 m and 1.0 m [32]. The Generalized Hoek-Brown criterion was used for the rock mass, and the Double-Yield model was used in the caving zone. The bolt and anchor cable were applied in the roadway. At the same time, the slap, unit support or portal support was also used. The right and left side of the model restrained the horizontal displacement, and the bottom was restrained the vertical displacement. 10 MPa uniform vertical loads were applied on the top of the model to replace the overlying strata weight above 400 m [30], and the ground stress was also applied.

Fig.8 The numerical model**4.2 The key parameter optimization of the roof-cutting technology****4.2.1 Roof-cutting height**

As shown in Figure 9, after the roof collapse, vertical stress states of the roadway surrounding rock were extracted under different roof cutting heights, and the anchor cable and bolt tensile stresses and the resistance of the unit support were monitored.

Fig. 9 Stress states of the roadway surrounding rock

Figure 9 indicates that:

(1) When the roof-cutting heights were 4 m, 6 m and 8 m, the axial tensions of the bolt were 126 kN, 91 kN and 73 kN, respectively; the axial tensions of the anchor cable were 156 kN, 187 kN and 285 kN, respectively; and the working resistances of the unit support were 44.3 MPa, 41.1 MPa and 35.2 MPa, respectively. After roof collapses, the roof lost constraint forces on one side of the roadway, and this forces were replaced by the suspension action of the anchor cable. Therefore, as the roof-cutting height increased, the tension of the anchor cable also increased. At this point, the strata were compacted under effect of anchor cables, so the tension of the bolt decreased. As the roof-cutting height increased, the roof thickness required to break decreases, so the working resistance for the unit support gradually reduced.

(2) When the roof-cutting heights were 4 m, 6 m and 8 m, the distances between positions of the stress peak in coal wall and the roadway ribs were 14.5 m, 18.2 m and 20.5 m, respectively, and the stresses were 24.6 MPa, 23.6 MPa and 21.2 MPa, respectively. With the increase of the roof-cutting heights, the stress concentration peak became smaller. The longer the distance between the stress concentration position to the ribs, the more stable the roadway would be.

(3) To further analyze the roadway surrounding rock failure conditions under different roof cutting heights, various intensity factors of the roadway surrounding rock were given. When the intensity factor was more than 1.0, the surrounding rock was damaged. The analysis indicated that the higher the roof-cutting height, the worse the surrounding rock failure.

The limited value of the working resistance of the unit support is 45 MPa, and the working resistance of 40MPa was chosen for safety in this study. Meanwhile, the tension of the anchor cable was 250 kN. Based on the above requirements, the cutting height was determined as 6 m.

Fig.10 Failure zones of the roadway surrounding rocks**4.2.2 Type of the temporary support equipment**

The temporary support equipment is mainly used to provide a certain support for the broken roof and assist to retain the roadway. The main temporary support equipment includes the single prop, unit support and portal support.

The single prop is low-cost and easy to move, but its working resistance is relatively small and the support effects are poor. The unit support and portal support could provide larger support. The former could provide support for a point while the latter could provide support for the whole roadway.

Fig.11 The temporary support equipment

Due to high support intensity of the unit support and portal support, their support effects were mainly compared. As shown in Figure 11, the roof-cutting height was 6 m in the numerical simulation. As shown in Figure 12, The surrounding rock vertical stress at 1 m away from the roadway roof, the lateral compressive stress of the slab and the lateral displacement of the caving gangues were extracted, respectively.

Fig.12 Support effects comparison of the unit support and portal support

As shown in Figure 12 (a), the vertical stress of the surrounding rock with 1.0 m distance to the roadway roof was extracted. The comparative analyses indicated that: (1) The support intensity of the portal support was higher than that of the unit support; (2) The unit support and portal support played a good role in supporting roof at the goaf side, so the vertical stress increased sharply; (3) The supporting effect of the portal support was obviously better than that of the unit support near the side of the solid coal.

As shown in Figure 12 (b), the lateral extrusion stresses to the slab were extracted. The comparative analyses indicated that the lateral stress was significantly smaller with the portal support. The lateral extrusion stress peaks corresponding to the unit support and that to portal support were 6.2 MPa and 4.6 MPa, respectively. Compared with the unit support, the portal support had higher working resistance and could share greater roof pressure, thereby reducing the vertical extrusion stress from roof to the gangues in the goaf and further lowering the lateral stresses due to gangue extrusion.

As shown in Figure 12 (c), the lateral displacements of the broken roof were extracted. The comparative analysis indicated that the lateral displacement was smaller with that of the portal support. The reason was the same as the lateral extrusion stress to slab from broken roof.

To further compare and analyze the failure conditions of the roadway surrounding rocks with different temporary support equipment, the failure areas were given under supporting by the unit support and portal support, as shown in Figure 13.

Fig.13 Roadway surrounding rock failure areas

Figure 13 showed that compared with the unit support, the failure height of the roadway roof was significantly smaller supported by the portal support, and the failure area on the left side of the roadway was slightly smaller. The failure area was more serious in the middle of the roadway roof supported by unit supports.

By analyzing the surrounding rock vertical stress, the lateral extrusion stress to the slab, the lateral displacement of the broken roof and failure areas of the surrounding rock, the supporting effects of the portal support were superior.

4.2.3 The working resistance of portal support

With the advantages of higher support strength and better effect on roadway deformation and failure control, the portal support was preferred for temporary support. The resistance was an important parameter. The support strength was directly related with the diameter of the hydraulic cylinder. The support resistance could be obtained by formula (12):

$$F = \frac{\pi}{2} D^2 P_0 \quad (12)$$

Fig.14 Roadway surrounding rock failure states under different working resistances

In general, the diameter of the portal support's oil cylinder was between 100 mm and 300 mm, and the working pressure P_0 was 40 MPa. When the diameters were 100 mm, 200 mm and 300 mm, the corresponding supporting forces were 628 kN, 2512 kN and 5652 kN, respectively. As shown in Figure 14, the failure areas of the roadway

325 surrounding rocks were given under three working resistance by the numerical simulation method. As shown in
 326 Figure 14, when the support was 628 kN, the failure area was relatively large. When the support was 2512 kN, the
 327 surrounding rock failure was reduced obviously. Therefore, the diameter of the portal support's oil cylinder should
 328 be above 200 mm.

329 **4.2.4 Parameters of the bolt and anchor cable**

330 As the active support, the prestress bolt support could control dilatancy and deformation in bolted surrounding
 331 rocks, such as the roof separation, sliding, fracture opening and new crack generation. They could keep the
 332 surrounding rocks under pressure, which could restrain the bending and deformation, tensile and shear failure,
 333 maintain the integrity of the surrounding rock and decrease the strength of the surrounding rock [33-35]. The pre-
 334 tightening force, density and length of the bolt and anchor cable need to be optimized at the design time. As shown
 335 in Figure 15, the FLAC was used to simulate and optimize these parameters according to the prestress theory.

336

337

337 **Fig.15 Parameters optimization of the bolt and anchor cable**

338 As shown in Figure 15, the continuous compressive stress areas were formed around the roadway surrounding
 339 rock with 80 kN pre-tightening force of the bolt, 0.8 m bolt spacing, 2.0 m length of the bolt and 4 anchor cables.
 340 These parameters were selected as the support parameters for the roadway.

341 **5 Engineering application**

342 **5.1 The industrial experiment**

343 In order to adapt underground roof-cutting process, the machine of roof cutting with chain arm was operated
 344 on the ground, as shown in Figure 16 (a). The skilled ground operation was conducive to master the cutting
 345 technology and provide basis for underground engineering application. As shown in Figure 16 (b), the machine was
 346 applied in the underground roadway to conduct roof-cutting experiment. The accumulated cutting length was 150
 347 m, and the average daily cutting progress was 3.5 m.

348

(a) Ground experiment

(b) Underground experiment

349

349 **Fig.16 The chain arm sawing roof experiment**

349 **5.2 The surrounding rock damage due to roof cutting**

350 The kerf was formed by the machine of roof cutting with chain arm whose influence on the mechanical
 351 properties of roof surrounding rocks was different from that of the common energizing blasting roof cutting. To
 352 study the influence of roof cutting damage of the roof surrounding rocks, the uniaxial compressive strength in-situ
 353 test system was used to compare the uniaxial compressive strength of part with the roof cutting and the part without
 354 being cut. To ensure the accuracy of the mechanical properties change, the monitoring points were arranged at the
 355 end of the kerf formed by the chain arm and is 20 mm away from the kerf. Figure 17 shows the comparison results.

356

357

357 **Fig.17 Change law of the uniaxial compressive strength of surrounding rocks**

358 Figure 17 indicated that: (1) The uniaxial compressive strength of the roof surrounding rocks with and that
 359 without roof cutting changed rarely, which was only around 5%. Therefore, the damage to surrounding rock by roof
 360 cutting with chain arm is small and the damage range to roof is about 82 mm; (2) In the range of bolt and anchor
 361 range of 0~2.5 m, the uniaxial compressive strength damage was less within the range of 0~0.6 MPa. It increased
 362 significantly beyond the bolt and anchor range of 2.5~6.0 m, and the highest was 3.37 MPa.

363 **6 Conclusions**

364 (1) The structure characteristics of the equipment of roof cutting with chain arm and the roof-cutting process
 365 were firstly illustrated. Then the technological process, roof-cutting parameters and damage ranges were given and

analyzed.

(2) The numerical calculation was used to optimize the roof-cutting height, the type of the temporary support equipment, working resistance of the portal support and the support parameters of the bolt and anchor cables. It was obtained that as the roof-cutting height increased, the tension of the roadway anchor cable increased, and the working resistance and the tension of the bolt decreased.

(3) The underground industrial experiment of roof cutting with chain arm was carried out. The damage of the surrounding rocks before and after roof cutting had little change. The variation range of the uniaxial compressive strength was only 5%, so the damage range to roof is about 82 mm.

Data Availability: All original data are deposited at Figshare: <https://figshare.com/s/c9a6404ed59ae8e66f62>.

Competing Interests: The authors declare no competing financial interests.

Authors' Contributions: B.Y participated in the design of the study; Y.T. drafted the manuscript; R.G. carried out the statistical analyses; Q.Y. collected field data; Z.L. conceived of the study and designed the study; H.X. coordinated the study and helped draft the manuscript. All authors gave final approval for publication.

Funding: This work was funded by the State Key Research Development Program of China (grant number 2018YFC0604500); Talents Project of Liaoning Revitalization (XLYC201807219).

Ethics: The article is about mining science and does not involve ethics.

Permission to carry out fieldwork: We obtained the appropriate permissions and licenses to conduct the fieldwork detailed in our study.

Acknowledgements: This work was funded by the State Key Research Development Program of China (grant number 2018YFC0604500); Talents Project of Liaoning Revitalization (XLYC201807219). The authors gratefully acknowledge the financial support from the organization.

References

- [1] Duncan, G., Sobey, G. & Clarke, T. 2007 Top coal caving longwall maximizes thick seam recovery - Astar's longwall system offers opportunities in seams thicker than 4.5 meters. *E&Mj-Engineering and Mining Journal* **208**, 50-56.
- [2] Zhu, W.B., Xu, J.M., Xu, J.L., Chen, D.Y. & Shi, J.X. 2017 Pier-column backfill mining technology for controlling surface subsidence. *International Journal of Rock Mechanics and Mining Sciences* **96**, 58-65. (doi:10.1016/j.ijrmms.2017.04.014).
- [3] Ma, Z., Wang, J., He, M., Gao, Y., Hu, J. & Wang, Q. 2018 Key Technologies and Application Test of an Innovative Noncoal Pillar Mining Approach: A Case Study. *Energies* **11**. (doi:10.3390/en11102853).
- [4] Wang, Q., Gao, H., Jiang, B., Li, S., He, M., Wang, D., Lu, W., Qin, Q., Gao, S. & Yu, H. 2017 Research on reasonable coal pillar width of roadway driven along goaf in deep mine. *Arab J Geosci* **10**. (doi:10.1007/s12517-017-3252-1).
- [5] Yang, J.X., Liu, C.Y., Yu, B. & Wu, F.F. 2014 The effect of a multi-gob, pier-type roof structure on coal pillar load-bearing capacity and stress distribution. *Bulletin of Engineering Geology and the Environment* **74**, 1267-1273. (doi:10.1007/s10064-014-0685-6).
- [6] Chang, J., Xie, G. & Yang, K. 2010 *Design of coal pillar with roadway driving along goaf in fully mechanized top-coal caving face* 235-241 p.
- [7] Sinha, S. & Walton, G. 2018 A progressive S-shaped yield criterion and its application to rock pillar behavior. *International Journal of Rock Mechanics and Mining Sciences* **105**, 98-109. (doi:10.1016/j.ijrmms.2018.03.014).
- [8] Zhang, C. & Tu, S. 2016 Control technology of direct passing karstic collapse pillar in longwall top-coal caving mining. *Natural Hazards* **84**, 17-34. (doi:10.1007/s11069-016-2402-1).
- [9] Feng, G., Wang, P. & Chugh, Y.P. 2018 Stability of Gate Roads Next to an Irregular Yield Pillar: A Case Study. *Rock Mechanics and Rock Engineering*. (doi:10.1007/s00603-018-1533-y).
- [10] Li, W., Bai, J., Peng, S., Wang, X. & Xu, Y. 2014 Numerical Modeling for Yield Pillar Design: A Case Study. *Rock Mechanics and Rock Engineering* **48**, 305-318. (doi:10.1007/s00603-013-0539-8).
- [11] Palei, S.K. & Das, S.K. 2009 Logistic regression model for prediction of roof fall risks in bord and pillar workings in coal mines: An approach. *Safety Science* **47**, 88-96. (doi:10.1016/j.ssci.2008.01.002).
- [12] He, M., Wang, Y., Yang, J., Zhou, P., Gao, Q. & Gao, Y. 2018 Comparative analysis on stress field distributions in roof cutting non-pillar mining method

- 1
2 411 and conventional mining method. *Journal of China Coal Society* **43**, 626–637.
- 3 412 [13] He, M., Li, C., Gong, W., Wang, J. & Tao, Z. 2016 Support principles of NPR bolts/cables and control techniques of large deformation. *Chinese Journal of*
4 413 *Rock Mechanics and Engineering* **35**, 1513-1529.
- 5 414 [14] Gao, Y., Yang, J., He, M., Wang, Y. & Gao, Q. 2017 Mechanism and control techniques for gangue rib deformations in gob-side entry retaining formed by
6 415 roof fracturing in thick coal seams. *Chinese Journal of Rock Mechanics and Engineering* **36**, 2492-2502.
- 7 416 [15] Gao, Y.-b., Guo, Z.-b., Yang, J., Wang, J.-w. & Wang, Y.-j. 2017 Steady analysis of gob-side entry retaining formed by roof fracturing and control techniques
8 417 by optimizing mine pressure. *Journal of China Coal Society* **42**, 1672-1681.
- 9 418 [16] Sun, X., Liu, X., Lliang, G., Wang, D. & Jiang, Y. 2014 Key parameters of gob-side entry retaining formed by roof cut and pressure releasing in thin coal
10 419 seams. *Chinese Journal of Rock Mechanics and Engineering* **33**, 1449-1456.
- 11 420 [17] Guo, Z., Wang, J., Cao, T., Chen, L. & Wang, J. 2016 Research on key parameters of gob-side entry retaining automatically formed by roof cutting and
12 421 pressure release in thin coal seam mining. *Journal of China University of Mining & Technology* **45**, 879-885.
- 13 422 [18] Ju, M.H., Li, X.H., Yao, Q.L., Li, D.W., Chong, Z.H. & Zhou, J. 2015 Numerical investigation into effect of rear barrier pillar on stress distribution around
14 423 a longwall face. *Journal of Central South University* **22**, 4372-4384. (doi:10.1007/s11771-015-2986-8).
- 15 424 [19] He, M., Gong, W., Wang, J., Qi, P., Tao, Z., Du, S. & Peng, Y. 2014 Development of a novel energy-absorbing bolt with extraordinarily large elongation
16 425 and constant resistance. *International Journal of Rock Mechanics and Mining Sciences* **67**, 29-42. (doi:10.1016/j.ijrmms.2014.01.007).
- 17 426 [20] Li, G., Wang, W., Jing, Z., Zuo, L., Wang, F. & Wei, Z. 2018 Mechanism and numerical analysis of cutting rock and soil by TBM cutting tools. *Tunnelling*
18 427 *and Underground Space Technology* **81**, 428-437. (doi:10.1016/j.tust.2018.08.015).
- 19 428 [21] Otto, A. & Parmigiani, J.P. 2018 Cutting performance comparison of low-kickback saw chain. *International Journal of Forest Engineering* **29**, 83-91.
20 429 (doi:10.1080/14942119.2018.1449444).
- 21 430 [22] Suchowerska, A.M., Merifield, R.S., Carter, J.P. & Clausen, J. 2012 Prediction of underground cavity roof collapse using the Hoek-Brown failure criterion.
22 431 *Computers and Geotechnics* **44**, 93-103. (doi:10.1016/j.compgeo.2012.03.014).
- 23 432 [23] Bertuzzi, R., Douglas, K. & Mostyn, G. 2016 Improving the GSI Hoek-Brown criterion relationships. *International Journal of Rock Mechanics and Mining*
24 433 *Sciences* **89**, 185-199. (doi:10.1016/j.ijrmms.2016.09.008).
- 25 434 [24] G, S.M.D. 1990 Mechanism of caving in longwall coal mining. In *Rock Mechanics Contributions and Challenges: Proceedings of*
26 435 *the 31st US Symposium* (pp. 161-168. Golden, Colorado.
- 27 436 [25] Zhang, G.C., He, F.L., Jia, H.G. & Lai, Y.H. 2017 Analysis of Gateroad Stability in Relation to Yield Pillar Size: A Case Study. *Rock Mechanics and Rock*
28 437 *Engineering* **50**, 1263-1278. (doi:10.1007/s00603-016-1155-1).
- 29 438 [26] Bai, Q.-S., Tu, S.-H., Zhang, X.-G., Zhang, C. & Yuan, Y. 2013 Numerical modeling on brittle failure of coal wall in longwall face—a case study. *Arab J*
30 439 *Geosci* **7**, 5067-5080. (doi:10.1007/s12517-013-1181-1).
- 31 440 [27] Bai, Q.S., Tu, S.H., Chen, M. & Zhang, C. 2016 Numerical modeling of coal wall spall in a longwall face. *International Journal of Rock Mechanics and*
32 441 *Mining Sciences* **88**, 242-253. (doi:10.1016/j.ijrmms.2016.07.031).
- 33 442 [28] Wang, P.F., Zhao, J.L., Chugh, Y.P. & Wang, Z.Q. 2017 A Novel Longwall Mining Layout Approach for Extraction of Deep Coal Deposits. *Minerals* **7**, 60.
34 443 (doi:ARTN 60
35 444 10.3390/min7040060).
- 36 445 [29] Bai, Q.S., Tu, S.H., Zhang, C. & Zhu, D.F. 2016 Discrete element modeling of progressive failure in a wide coal roadway from water-rich roofs. *International*
37 446 *Journal of Coal Geology* **167**, 215-229. (doi:10.1016/j.coal.2016.10.010).
- 38 447 [30] Li, X., Ju, M., Yao, Q., Zhou, J. & Chong, Z. 2015 Numerical Investigation of the Effect of the Location of Critical Rock Block Fracture on Crack Evolution
39 448 in a Gob-side Filling Wall. *Rock Mechanics and Rock Engineering* **49**, 1041-1058. (doi:10.1007/s00603-015-0783-1).
- 40 449 [31] Yang, S.-Q., Chen, M., Jing, H.-W., Chen, K.-F. & Meng, B. 2017 A case study on large deformation failure mechanism of deep soft rock roadway in Xin'An
41 450 coal mine, China. *Engineering Geology* **217**, 89-101. (doi:10.1016/j.enggeo.2016.12.012).
- 42 451 [32] Shabanimashcool, M. & Li, C.C. 2012 Numerical modelling of longwall mining and stability analysis of the gates in a coal mine. *International Journal of*
43 452 *Rock Mechanics and Mining Sciences* **51**, 24-34. (doi:10.1016/j.ijrmms.2012.02.002).
- 44 453 [33] Zheng, X., Feng, X., Zhang, N., Gong, L. & Hua, J. 2014 Serial decoupling of bolts in coal mine roadway supports. *Arab J Geosci* **8**, 6709-6722.
45 454 (doi:10.1007/s12517-014-1697-z).
- 46 455 [34] Ma, S., Nemic, J., Aziz, N. & Zhang, Z. 2014 Analytical model for rock bolts reaching free end slip. *Construction and Building Materials* **57**, 30-37.
47 456 (doi:10.1016/j.conbuildmat.2014.01.057).

1
2
3
4
5
6
7
8
9
10
11
12
13
14
15
16
17
18
19
20
21
22
23
24
25
26
27
28
29
30
31
32
33
34
35
36
37
38
39
40
41
42
43
44
45
46
47
48
49
50
51
52
53
54
55
56
57
58
59
60

457 [35] Pan, R., Wang, Q., Jiang, B., Li, S.C., Sun, H.B., Qin, Q., Yu, H.C. & Lu, W. 2017 Failure of bolt support and experimental study on the parameters of bolt-
458 grouting for supporting the roadways in deep coal seam. *Eng Fail Anal* **80**, 218-233. (doi:10.1016/j.engfailanal.2017.06.025).

Figure caption lists

- 1
2
3 460
4
5 461 Fig.1 Layout method of coal pillars. (a) Position of the roadway; (b) Stress state of the roadway
6
7
8 462 Fig.2 The roadway failure areas in the small pillar method
9
10 463 Fig.3 The treatment methods for roadway. (a) The traditional method; (b) Non-pillar method
11
12
13 464 Fig.4 The machine of roof cutting with chainsaw arm
14
15 465 Fig.5 The process of roof cutting with chainsaw arm
16
17
18 466 Fig.6 The process of retaining roadway
19
20 467 Fig.7 Working conditions of the temporary support equipment
21
22
23 468 Fig.8 The numerical model
24
25 469 Fig. 9 Stress states of the roadway surrounding rock. (a) 4 m; (b) 6 m; (c) 8 m
26
27
28 470 Fig.10 Failure zones of the roadway surrounding rocks
29
30 471 Fig.11 The temporary support equipment
31
32
33 472 Fig.12 Support effects comparison of the unit support and portal support. (a)The vertical stress; (b)
34
35 473 The lateral compressive stress to the slab; (c) The lateral displacement of the gangues
36
37 474 Fig.13 Roadway surrounding rock failure areas
38
39
40 475 Fig.14 Roadway surrounding rock failure states under different working resistances
41
42 476 Fig.15 Parameters optimization of the bolt and anchor cable. (a) The pre-tightening force of the
43
44 477 bolt; (b) The length of the bolt; (c) The density of the bolt;(d) The density of the anchor cable
45
46 478 Fig.16 The chainsaw arm sawing roof experiment (a) Ground experiment; (b) Underground
47
48 479 experiment
49
50
51 480 Fig.17 Comparison of the uniaxial compressive strength of surrounding rocks
52
53
54
55
56
57
58
59
60

1
2
3
4
5
6
7
8
9
10
11
12
13
14
15
16
17
18
19
20
21
22
23
24
25
26
27
28
29
30
31
32
33
34
35
36
37
38
39
40
41
42
43
44
45
46
47
48
49
50
51
52
53
54
55
56
57
58
59
60

Table caption lists

481
482
483
484
485
486

Table 1 Coefficients for the caving zone height
Table 2 Working resistance of the temporary support equipment
Table 3 Mechanical properties of the supporting materials

The Figures

487

(a) Position of the roadway

(b) Stress state of the roadway

488

Fig.1 Layout method of coal pillars

Fig.2 The roadway failure areas in the small pillar method

489

490

491

(a) The traditional method

(b) Non-pillar method

Fig.3 The treatment methods for roadway

492

Fig.4 The machine of roof cutting with chainsaw arm

Fig.5 The process of roof cutting with chainsaw arm

Fig.6 The process of retaining roadway

Fig.7 Working conditions of the temporary support equipment

Fig.8 The numerical model

Fig.9 Stress states of the roadway surrounding rock

Fig.10 Failure zones of the roadway surrounding rocks

Fig.11 The temporary support equipment

Fig.12 Support effects comparison of the unit support and portal support

Fig.13 Roadway surrounding rock failure areas

Fig.14 Roadway surrounding rock failure states under different working resistances

(a) The pre-tightening force of the bolt

(b) The length of the bolt

(c) The density of the bolt

(d) The density of the anchor cable

Fig.15 Parameters optimization of the bolt and anchor cable

(b) Ground experiment

(b) Underground experiment

Fig.16 The chainsaw arm sawing roof experiment

Fig.17 Comparison of the uniaxial compressive strength of surrounding rocks

The Tables

Table 1 Coefficients for the caving zone height

Type of the immediate roof	Uniaxial compressive strength /MPa	Coefficients	
		c_1	c_2

Hard	>40	2.1	16
Mid-hard	20~40	4.7	19
Weak	<20	6.2	32

533

534

Table 2 Working resistance of the temporary support equipment (He et al. 2014)

Broken characteristics of the strata	Mechanical model	Working resistance
		$P = [M_B(F_G / K_G + x_0 + a + b)K_G / F_G + q(F_G / K_G)^2 + q(x_0 + a + b)^2 / 2 + qF_G(x_0 + a + b) / 2 / K_G + q_0(x_0 + a + b)^2 / 2 - F_G(x_0 + a + b) - qL(h_0 - \Delta S_C) / [4(h - \Delta S_C)] - M_A - M_0 - \int_0^{x_0} \sigma(x_0 - x)dx] / (x_0 + a + b / 2)$

535

536

537

538

539

540

541

542

543

544

Remarks: ΔS_C is the settlement of the rock block C at C' , m; T_B is the lateral horizontal force of the rock block B at B' , kN; T_C is the lateral horizontal force of the rock block C at C' , kN; N_B and N_C are the shear stresses of the rock block B and C, kN; σ is the supporting force for the roof from the coal mass in the plastic zone, MPa; q is the weight from the roof and the average load of its overlying weak strata, kN/m; q_0 is the average load of the immediate roof, kN/m; M_A and M_B are residual moments of the rock beam B at A' and B' , kN·m; M_0 is the bending moment of the immediate roof to the roof, kN·m; K_G is the support coefficient of the gangues in the goaf, kN/m; F_G is the support stress of the goaf to the roof, kN/m; h_0 is the thickness of the roof, m; P is the support resistance, kN; x_0 is the width of the limit equilibrium zone of the lateral coal mass, m; a is the width of the roadway retained, m; b is the width of the temporary support equipment, m.

Table 3 Mechanical properties of the supporting materials

Contact attributes	Values
Bolt/anchor cable	
Elastic modulus/GPa	210
Yield strength/kN	250/470
Pre-tightening force/kN	100/150
Stiffness of the resin roll /N/m/m	2e9
Cohesive force of the resin roll /N/m	4e5
W-type steel strip and metal net	
Elastic modulus/GPa	210
Tensile strength/MPa	500
Compressive strength/MPa	500
Normal stiffness of the interface/GPa/m	10
Shear stiffness of the interface/GPa/m	10

545

546

547

Appendix B

Cover letter

Dear editor

Thank you for giving me the chance to revise my manuscript. All revisions are marked with red and blue fonts in the article. If you have any questions, please contact me.

Yours sincerely,

Yang Tai

E-mail: cumtcqty@163.com

The Revision for Comments

Reviewer: 1

1. The manuscript requires some solid editing by a native English speaker. Articles are missing, familiar language inappropriate for scientific journals must be corrected.

Thank you for your comments. Articles are language-polished by professional institutions and marked with red font.

2. Some mention of the findings should be found in the abstract. Simply saying something is very good is general, ambiguous and of little value to the reader.

Thank you for your comments. The abstract of the article has been rewritten as follows:

China consumes more than 3.6 billion tons of coal every year, accounting for over 50% of the global coal consumption. Therefore, the sustainable development of coal mine is a problem needed to be solved by the Chinese government. During the coal resources recovery, the protective coal pillars between the adjacent working faces cause the serious loss for coal resources. In order to solve the problem, it was put forward that the new technology of roof cutting with chain arm to retain roadway in the paper. Firstly, the process of retaining roadway, roof-cutting parameters and the damage ranges of roadway surrounding rock induced by roof cutting with chain arm were analysed. Then, it was given that the working resistance of the temporary support equipment in technology of roof cutting with chain arm to retain roadway. Next, the roof-cutting height, the type of temporary support equipment, working resistance of portal support and support parameters of the bolt and anchor cables were optimized by the numerical calculation. Finally, the industrial experiment of retaining roadway by roof cutting with chain arm was carried out in a working face. The surrounding rock damage was minimal with the use of chain arm-roof cutting technology, and the variation range of the uniaxial compressive strength was only 5%, resulting in the roof damage rate to 82 mm. From the studies, it was concluded that this technology could be of a great asset to the coal mining community.

3. The authors have relied almost exclusively on Chinese literature. I am certain there are more scholars globally.

Thank you for your comments. Because the research on coal mining is mainly concentrated in China, the most of references in this article is the papers, which is about Chinese research status published by Chinese scholars and indexed by SCI. In order to further increase the internationality of the article, the article also adds some articles by foreign scholars, as follows:

Prasetyo, S.H., Irnawan, M.A., Simangunsong, G.M., Wattimena, R.K., Arif, I. & Rai, M.A. 2019 New coal pillar strength formulae considering the effect of interface friction. *International Journal of Rock Mechanics and Mining Sciences* 123. (doi: 10.1016/j.ijrmms.2019.104102).

Sinha, S. & Walton, G. 2020 Modeling behaviors of a coal pillar rib using the progressive S-shaped yield criterion. *Journal of Rock Mechanics and Geotechnical Engineering*. (doi: 10.1016/j.jrmge.2019.12.002).

Garza-Cruz, T., Pierce, M. & Board, M. 2019 Effect of Shear Stresses on Pillar Stability: A Back Analysis of the Troy Mine Experience to Predict Pillar Performance at Montanore Mine. *Rock Mechanics and Rock Engineering* 52, 4979-4996. (doi:10.1007/s00603-019-02011-3).

Reviewer: 2

1. It was obtained that the surrounding rock damage changed slightly before and after roof cutting, and the variation range of the uniaxial compressive strength was only around 5%. The effect of cutting roof and retaining roadway is very good. — Rewrite this part.

Thank you for your comments. The article is modified as follows:

The surrounding rock damage was minimal with the use of chain arm-roof cutting technology, and the variation range of the uniaxial compressive strength was only 5%, resulting in the roof damage rate to 82 mm. From the studies, it was concluded that this technology could be of a great asset to the coal mining community.

2. According to the width for the coal pillars, it can be divided into large-pillar method and small-pillar method. — Rewrite this part.

Thank you for your comments. The article is modified as follows:

Based on the width of the coal pillars, the design method of protective coal pillars can be divided into large-pillar method and small-pillar method.

3. Use one term. Either goaf or gob in the whole paper.

Thank you for your comments. The article is modified as follows: The goaf is unified to gob in the article.

4. In order to reduce the loss of coal pillars and improve coal recovery rate in coal mines — Rewrite this part.

Thank you for your comments. The article is modified as follows:

In order to reduce the size of coal pillars and improve coal recovery rate in coal mines

5. This method could make the roadway in the stress reduction area where the roadway is in an easy maintenance and the loss of coal resources could be effectively avoided. — Rewrite this part.

Thank you for your comments. The article is modified as follows:

By applying this method, the roadway will locate in the stress reduction area, which make the maintenance of the roadway easier and increase the recovery rate of coal resource.

6. use "studies" instead of researches.

Thank you for your comments. The article is modified as follows: The 'researches' is modified to the 'studies' in the article.

7. The aforementioned researches mainly focus on non-pillar mining method to retain goaf-side roadway from the aspect of theory and technology. And the energizing blasting technology is usually adopted as the roof-cutting method. — Rewrite this part.

Thank you for your comments. The article is modified as follows:

The aforementioned studies mainly focus on non-pillar mining method to retain gob-side roadway. And the energizing blasting method is usually adopted as the roof-cutting method.

8. What' s more, blasting is prohibited in mines with high gas. — Rewrite this part.

Thank you for your comments. The article is modified as follows:

In some extreme conditions like high gassy coal mine, blasting is prohibited.

9. Throughout the paper I have noticed etc. in several places. Please avoid this and mention them in particular. Etc. is not suggested in a technical paper.

Thank you for your comments. The article is modified as follows: The 'etc' is modified to the 'and so on'.

10. after —After

Thank you for your comments. The article is modified as follows:

After the working face pass, temporary supports in the roadway should be strengthened

11. roadway temporary supports

Thank you for your comments. The article is modified as follows: After the working face pass, temporary supports in the roadway should be strengthened

12. The structure characteristics of the equipment of roof cutting with chain arm and the roof-cutting process were firstly illustrated. Then the technological process, roof-cutting parameters and damage ranges were given and analysed. —— Not a conclusive statement. What's achieved and the punchy outcome from the study are conclusions.

Thank you for your comments. The article is modified as follows:

Based on theoretical analysis, it was given that the working resistance of the temporary support equipment in technology of roof cutting with chain arm to retain roadway.

13. obtained is not the correct term here. Use something like "observed". Overall this sentence needs revision.

Thank you for your comments. The article is modified as follows:

It was observed that no obvious surrounding rock damage changed slightly before and after roof cutting

14. The underground industrial experiment of roof cutting with chain arm was carried out. The damage of the surrounding rocks before and after roof cutting had little change. The variation range of the uniaxial compressive strength was only 5%, so the damage range to roof is about 82 mm. —— Revision needed.

Thank you for your comments. The article is modified as follows:

It was observed that no obvious surrounding rock damage changed slightly before and after roof cutting, and the variation fluctuation range of the uniaxial compressive strength was only around 5%. Therefore, the effect of cutting the roof and retaining the roadway is very good have proved its promising results.

Appendix C

Dear editor

Thank you for accepting the paper and me the chance to revise my manuscript. The language has been improved. Besides, the text file of the manuscript, a separate electronic file, a 100-word media summary, raw data etc. have been prepared. If you have any question, please contact me.

Yours sincerely,

Yang Tai

E-mail: cumtcqy@163.com